# Confidence-Aware Imitation Learning
# from Demonstrations with Varying Optimality

**Songyuan Zhang**[1][*], **Zhangjie Cao**[2][*], **Dorsa Sadigh**[2], **Yanan Sui**[1]

[1]National Engineering Lab for Neuromodulation, SAE, Tsinghua University, China
[2]Department of Computer Science, Stanford University, USA
szhang21@mit.edu, {caozj,dorsa}@cs.stanford.edu, ysui@tsinghua.edu.cn

## Abstract

Most existing imitation learning approaches assume the demonstrations are drawn from experts who are optimal, but relaxing this assumption enables us to use a wider range of data. Standard imitation learning may learn a suboptimal policy from demonstrations with varying optimality. Prior works use confidence scores or rankings to capture beneficial information from demonstrations with varying optimality, but they suffer from many limitations, *e.g.*, manually annotated confidence scores or high average optimality of demonstrations. In this paper, we propose a general framework to learn from demonstrations with varying optimality that jointly learns the confidence score and a well-performing policy. Our approach, Confidence-Aware Imitation Learning (CAIL) learns a well-performing policy from confidence-reweighted demonstrations, while using an outer loss to track the performance of our model and to learn the confidence. We provide theoretical guarantees on the convergence of CAIL and evaluate its performance in both simulated and real robot experiments. Our results show that CAIL significantly outperforms other imitation learning methods from demonstrations with varying optimality. We further show that even without access to any optimal demonstrations, CAIL can still learn a successful policy, and outperforms prior work.

## 1 Introduction

We consider an imitation learning setting that learns a well-performing policy from a mixture of demonstrations with varying optimality ranging from random trajectories to optimal demonstrations. As opposed to standard imitation learning, where the demonstrations come from experts and thus are optimal, this benefits from a larger and more diverse source of data. Note that different from setting that the demonstrations are optimal but lack some causal factors [31], in our setting, the demonstrations can be suboptimal. However, this introduces a new set of challenges. First, one needs to select useful demonstrations beyond the optimal ones. We are interested in settings where we do not have sufficient expert demonstrations in the mixture so we have to rely on learning from sub-optimal demonstrations that can still be successful at parts of the task. Second, we need to be able to filter the negative effects of useless or even malicious demonstrations, *e.g.*, demonstrations that implicitly fail the tasks.

To address the above challenges, we propose to use a measure of *confidence* to indicate the likelihood that a demonstration is optimal. A confidence score can provide a fine-grained characterization of each demonstration's optimality. For example, it can differentiate between near-optimal demonstrations or adversarial ones. By reweighting demonstrations with a confidence score, we can simultaneously learn from useful but sub-optimal demonstrations while avoiding the negative effects of malicious ones. So

---

[*]Equal contribution.

35th Conference on Neural Information Processing Systems (NeurIPS 2021).

our problem reduces to learning an accurate confidence measure for demonstrations. Previous work learns the confidence from manually annotated demonstrations [30], which are difficult to obtain and might contain bias—For example, a conservative and careful demonstrator may assign lower confidence compared to an optimistic demonstrator to the same demonstration. In this paper, we remove restrictive assumptions on the confidence, and propose an approach that automatically learns the confidence score for each demonstration based on evaluation of the outcome of imitation learning. This evaluation often requires access to limited *evaluation data*.

We propose a new algorithm, Confidence-Aware Imitation Learning (CAIL), to jointly learn a well-performing policy and the confidence for every state-action pair in the demonstrations. Specifically, our method adopts a standard imitation learning algorithm and evaluates its performance to update the confidence scores with an evaluation loss, which we refer to as the *outer loss*. In our implementation, we use a limited amount of ranked demonstrations as our evaluation data for the outer loss. We then update the policy parameters using the loss of the imitation learning algorithm over the demonstrations reweighted by the confidence, which we refer to as the *inner loss*. Our framework can accommodate any imitation learning algorithm accompanied with an evaluation loss to assess the learned policy.

We optimize for the inner and outer loss using a bi-level optimization [5], and prove that our algorithm converges to the optimal confidence assignments under mild assumptions. We further implement the framework using Adversarial Inverse Reinforcement Learning (AIRL) [14] as the underlying imitation learning algorithm along with its corresponding learning loss as our inner loss. We design a ranking loss as the outer loss, which is compatible with the AIRL model and only requires easy-to-access ranking annotations rather than the exact confidence values.

The main contributions of the paper can be summarized as:

- We propose a novel framework, Confidence-Aware Imitation Learning (CAIL), that jointly learns confidence scores and a well-performing policy from demonstrations with varying optimality.

- We formulate our problem as a modified bi-level optimization with a pseudo-update step and prove that the confidence learned by CAIL converges to the optimal confidence in $\mathcal{O}(1/\sqrt{T})$ ($T$ is the number of steps) under some mild assumptions.

- We conduct experiments on several simulation and robot environments. Our results suggest that the learned confidence can accurately characterize the optimality of demonstrations, and that the learned policy achieves higher expected return compared to other imitation learning approaches.

## 2   Related Work

**Imitation Learning.** The most common approaches for imitation learning are Behavioral Cloning (BC) [20, 4, 23, 22, 3], which treats the problem as a supervised learning problem, and Inverse Reinforcement Learning (IRL), which recovers the reward function from expert demonstrations and finds the optimal policy through reinforcement learning over the learned reward  [1, 21, 32]. More recently, Generative Adversarial Imitation Learning (GAIL) [18] learns the policy by matching the occupancy measure between demonstrations and the policy in an adversarial manner [15]. Adversarial Inverse Reinforcement Learning (AIRL) [14] and some other approaches [13, 17] improve upon GAIL by simultaneously learning the reward function, and the optimal policy. However, these approaches assume that all the demonstrations are expert demonstrations, and cannot learn a well-performing policy when learning from demonstrations with varying optimality.

**Learning from Demonstrations with Varying Optimality: Ranking-based.** Ranking-based methods learn a policy from a sequence of demonstrations annotated with rankings [2, 25, 29, 10]. T-REX learns a reward from the ranking of the demonstrations and learns a policy using reinforcement learning [8]. In our work, we assume access to rankings of a small subset of the demonstrations. The reward function learned from such a small number of rankings by T-REX may have low generalization ability to out of distribution states. D-REX improves T-REX by automatically generating the rankings of demonstrations [9], and SSRR further finds the structure of the reward function [12]. These techniques automatically generate rankings under the assumption that a perturbed demonstration will have a lower reward than the original demonstration, which is not necessarily true for random or malicious demonstrations that can be present in our mixture. DPS utilizes partial orders and pairwise comparisons over trajectories to learn and generate new policies [19]. However, it requires interactively collecting feedback, which is not feasible in our offline learning setting.

**Learning from Demonstrations with Varying Optimality: Confidence-based.** Confidence-based methods assume each demonstration or demonstrator holds a confidence value indicating their optimality and then reweight the demonstrations based on this value for imitation learning. To learn the confidence, 2IWIL requires access to ground-truth confidence values for the demonstrations to accurately learn a confidence predictor [30]. Tangkaratt *et al.* require that all the actions for a demonstration are drawn from the same noisy distribution with sufficiently small variance [27]. IC-GAIL implicitly learns the confidence score by aligning the occupancy measure of the learned policy with the expert policy, but requires a set of ground-truth labels to estimate the average confidence [30]. Following works relax the assumption of access to the ground-truth confidence, but still require more optimal demonstrations than non-optimal ones in the dataset [26]. Other works require access to the reward of each demonstration [11]. All of these methods either rely on a specific imitation learning algorithm or require strong assumptions on the confidence. To move forward, we propose a general framework to jointly learn the confidence and the policy. Our framework is flexible as it can use any imitation learning algorithm as long as there exists a compatible outer loss, *i.e.,* the outer loss can evaluate the quality of the imitation learning model.

## 3  Problem Setting

We formulate the problem of learning from demonstrations with varying optimality as a Markov decision process (MDP): $\mathcal{M} = \langle \mathcal{S}, \mathcal{A}, \mathcal{T}, \mathcal{R}, \rho_0, \gamma \rangle$, where $\mathcal{S}$ is the state space, $\mathcal{A}$ is the action space, $\mathcal{T} : \mathcal{S} \times \mathcal{A} \times \mathcal{S} \to [0, 1]$ is the transition probability, $\rho_0$ is the distribution of initial states, $\mathcal{R} : \mathcal{S} \times \mathcal{A} \to \mathbb{R}$ is the reward function, and $\gamma$ is the discount factor. A policy $\pi : \mathcal{S} \times \mathcal{A} \to [0, 1]$ defines a probability distribution over the action space in a given state. The expected return, which evaluates the quality of a policy, can be defined as $\eta_\pi = \mathbb{E}_{s_0 \sim \rho_0, \pi} [\sum_{t=0}^{\infty} \gamma^t \mathcal{R}(s_t, a_t)]$, where $t$ indicates the time step.

We aim to learn a policy that imitates the behavior of a demonstrator $d$ following policy $\pi^d$ who provides a set of demonstrations $\Xi = \{\xi_1, \ldots, \xi_D\}$ and $\xi_i \sim \pi^d$. Each trajectory is a sequence of state-action pairs $\xi = \{s_0, a_0, \ldots, s_N\}$, and the expected return of a trajectory is $\eta_\xi = \sum_{t=0}^{N-1} \gamma^t \mathcal{R}(s_t, a_t)$.

A common assumption in classical imitation learning work is that the demonstrations are drawn from the expert policy $\pi^d = \pi^*$, *i.e.,* , the policy that optimizes the expected return of the MDP $\mathcal{M}$ [18, 14]. Here, we relax this assumption so that the demonstrations may contain non-expert demonstrations or even failures—drawn from policies other than $\pi^*$. Given the demonstration set $\mathcal{D}$, we need to assess our *confidence* in each demonstration. To achieve learning confidence over this mixture of demonstrations, we rely on the ability to evaluate the performance of imitation learning. This can be achieved by using an evaluation loss trained on *evaluation data*, $\mathcal{D}_E$ (as shown in Fig. 1). In our implementation, we rely on a small amount of rankings between trajectories as our evaluation data: $\mathcal{D}_E = \eta_{\xi_1} \geq \cdots \geq \eta_{\xi_m}$. To summarize, our framework takes a set of demonstrations with varying optimality $\mathcal{D}$ as well as a limited amount of evaluation data $\mathcal{D}_E$ along with an evaluation loss to find a well-performing policy. Note that unlike prior work [30], we do not assume that optimal demonstrations always exist in the demonstration set, and CAIL can still extract useful information from $\mathcal{D}$ while avoiding negative effects of non-optimal demonstrations.

## 4  Confidence-Aware Imitation Learning

In our framework, we adopt an imitation learning algorithm with a model $F_\theta$ parameterized by $\theta$ and a corresponding imitation learning loss $\mathcal{L}_{\text{in}}$, which we refer to as inner loss (as shown in Figure 1). We assign each state-action pair a confidence value indicating the likelihood of the state-action pair appearing in the well-performing policy. The confidence can be defined as a function mapping from a state-action pair to a scalar value $\beta : \mathcal{S} \times \mathcal{A} \to \mathbb{R}$. We aim to find the optimal confidence assignments $\beta^*$ to reweight state-action pairs within the demonstrations. We then conduct imitation learning from the reweighted demonstrations using the inner imitation loss $\mathcal{L}_{\text{in}}$ to learn a well-performing policy. Here, we first define the optimal confidence $\beta^*$ and describe how to learn it automatically.

**Defining the Optimal Confidence.** We define the distribution of state-action pairs visited by a policy $\pi$ based on the occupancy measure $\rho_\pi : \mathcal{S} \times \mathcal{A} \to \mathbb{R}$: $\rho_\pi(s, a) = \pi(a|s) \sum_{t=0}^{\infty} \gamma^t P(s_t = s|\pi)$, which can be explained as the un-normalized distribution of state transitions that an agent encounters when navigating the environment with the policy $\pi$. We can normalize the occupancy measure to form the

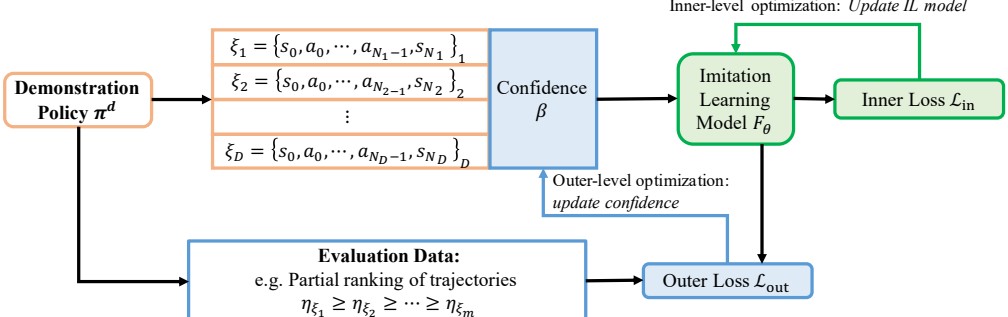

Figure 1: Confidence-Aware Imitation Learning. The demonstrations are shown in the orange box drawn from demonstration policy: $\xi_1, \ldots, \xi_D \sim \pi^d$. The confidence learning component and the outer loss are shown in blue. The confidence $\beta$ reweights the distribution of state-action pairs in the demonstration set, and then the imitation learning model $F_\theta$ learns a well-performing policy and new parameters $\theta$ with the confidence-reweighted distribution using the inner loss (imitation loss) shown in green. Next iteration, the updated $F_\theta$ generates new trajectories that are then evaluated by the outer loss and potentially other evaluation data (*e.g.* partial ranking of trajectories) to update confidence.

state-action distribution: $p_\pi(s,a) = \frac{\rho_\pi(s,a)}{\sum_{s,a} \rho_\pi(s,a)}$. Recall that $\pi^d$ is the policy that the demonstrations are derived from, which can potentially be a mixture of different expert, suboptimal, or even malicious policies. We reweight the state-action distribution of the demonstrations to derive a new state-action distribution, which corresponds to another policy $\pi_{\text{new}}$: $p_{\pi_{\text{new}}}(s,a) = \beta(s,a)p_{\pi^d}(s,a)$. Our goal is to find the optimal confidence $\beta^*$ that ensures the derived policy $\pi_{\text{new}}$ maximizes the expected return:

$$\beta^*(s,a) = \arg \max_\beta \eta_{\pi_{\text{new}}}. \tag{1}$$

With such $\beta^*(s,a)$, we can conduct imitation learning from the reweighted demonstrations to maximize the expected return with the provided demonstrations.

**Learning the Confidence.** We will learn an estimate of the confidence score $\beta$ without access to any annotations of the ground-truth values based on optimizing two loss functions: The inner loss and the outer loss. The inner loss $\mathcal{L}_{\text{in}}$ is accompanied with the imitation learning algorithm encouraging imitation, while the outer loss $\mathcal{L}_{\text{out}}$ captures the quality of imitation learning, and thus optimizing it finds the confidence value that maximizes the performance of the imitation learning algorithm.

Specifically, we first learn the imitation learning model parameters $\theta^*$ that minimize the inner loss:

$$\theta^*(\beta) = \arg \min_\theta \mathbb{E}_{(s,a) \sim \beta(s,a)p_{\pi^d(s,a)}} \mathcal{L}_{\text{in}}(s,a;\theta,\beta) \tag{2}$$

We note that the inner loss $\mathcal{L}_{\text{in}}(s,a;\theta,\beta)$ refers to settings where $(s,a)$ is sampled from the distribution $\beta(s,a)p_{\pi^d(s,a)}$, and hence implicitly depends on $\beta$. Thus we need to find the optimal $\beta^*$, which can be estimated by minimizing an outer loss $\mathcal{L}_{\text{out}}$:

$$\beta_{\text{out}}^* = \arg \min_\beta \mathcal{L}_{\text{out}}(\theta^*(\beta)). \tag{3}$$

This evaluates the performance of the underlying imitation learning algorithm with respect to the reward with limited evaluation data $\mathcal{D}_E$ (*e.g.* limited rankings if we select a ranking loss as our choice of $\mathcal{L}_{\text{out}}$; which we will discuss in detail in Sec. 4.3).

## 4.1 Optimization of Outer and Inner Loss

We design a bi-level optimization process consisting of an inner-level optimization and an outer-level optimization to simultaneously update the confidence $\beta$ and the model parameters $\theta$. Within the outer-level optimization, we first pseudo-update the imitation learning parameters to build a connection between $\beta$ and the optimized parameters $\theta'$ with the current $\beta$. We then update $\beta$ to make the induced $\theta'$ minimize the outer loss $\mathcal{L}_{\text{out}}$ in Eqn. (3). The inner-level optimization is to find the imitation learning model parameters that minimize inner loss $\mathcal{L}_{\text{in}}$ with respect to the confidence $\beta$. We introduce the details of the optimization below. We use $\tau$ to denote the number of iterations. Note that the losses in this section are all computed based on the expectation over states and actions.

**Outer-Level Optimization: Updating $\beta$.** Let $\beta_\tau$ be the confidence at time $\tau$. Using $\beta_\tau$, we first pseudo-update the imitation learning parameters $\theta$ using gradient descent. Here pseudo-update means that the update aims to compute the gradients of $\beta$ but does not really change the value of $\theta$. Let $\theta_0' = \theta_\tau$ be the current imitation learning model parameters, and we update $\theta'$ as:

$$\theta_{t+1}' = \theta_t' - \mu \nabla_{\theta'} \mathcal{L}_{\text{in}}(s, a; \theta_t', \beta_\tau), \tag{4}$$

where $\mu$ is the learning rate, $t$ is the pseudo-updating time step for $\theta'$. We will update $\theta'$ with respect to the fixed $\beta_\tau$ after convergence of Eqn. (4). After updating $\theta'$, we now update $\beta$ using gradient descent with the outer loss $\mathcal{L}_{\text{out}}$ from Eqn. (3):

$$\beta_{\tau+1} = \beta_\tau - \alpha \nabla_\beta \mathcal{L}_{\text{out}}(\theta'), \tag{5}$$

where $\alpha$ is the learning rate for updating $\beta$. Intuitively, updating $\beta$ as in Eqn. (5) aims to find the fastest update direction of $\theta'$ for decreasing the outer loss $\mathcal{L}_{\text{out}}$. Though we compute gradients of gradients for $\beta$ here, $\beta$ is only a one-dimension scalar for each state-action pair and within each iteration of training, we only sample a mini-batch of thousands of state-pairs for update. Thus, within each iteration, the total dimension of $\beta$ is small and computing the gradient of gradient is not costly.

**Inner-Level Optimization: Updating $\theta$.** With the updated $\beta_{\tau+1}$, we now will update $\theta$ using gradient descent, where we denote the initialization as $\theta_0 = \theta_\tau$.

$$\theta_{t+1} = \theta_t - \mu \nabla_\theta \mathcal{L}_{\text{in}}(s, a; \theta_t, \beta_{\tau+1}). \tag{6}$$

After convergence, we set $\theta_{\tau+1} = \theta$. With the two updates introduced above (outer and inner optimization), we finish one update iteration with setting $\beta_\tau$ to $\beta_{\tau+1}$ using the converged value from Eqn. (5) and $\theta_\tau$ to $\theta_{\tau+1}$ using the converged value from Eqn. (6).

In each iteration of the above optimization—in the steps of pseudo-updating and the steps of updating the imitation learning model—multiple gradient steps are required for convergence, meaning that there is a nested loop of gradient descent algorithms. The nested loop costs quadratic time and is inefficient especially for deep networks. To further accelerate the optimization, we propose an approximation, which only updates $\theta$ once in the pseudo-updating and the updating steps. Therefore, the new updating rule can be formalized as follows:

$$\begin{aligned}
\theta_{\tau+1}' &= \theta_\tau - \mu \nabla_\theta \mathcal{L}_{\text{in}}(s, a; \theta_\tau, \beta_\tau), \\
\beta_{\tau+1} &= \beta_\tau - \alpha \nabla_\beta \mathcal{L}_{\text{out}}(\theta_{\tau+1}'), \\
\theta_{\tau+1} &= \theta_\tau - \mu \nabla_\theta \mathcal{L}_{\text{in}}(s, a; \theta_\tau, \beta_{\tau+1}).
\end{aligned} \tag{7}$$

### 4.2 Theoretical Results

We analyze the convergence of the proposed bi-level optimization algorithm for the CAIL framework and derive the following theorems. We provide the detailed proofs of these theorems in Appendix.

**Theorem 1.** *(Convergence) Suppose the outer loss $\mathcal{L}_{out}$ is Lipschitz-smooth with constant L, the inequality*

$$\nabla_\theta \mathcal{L}_{out}(\theta_{\tau+1})^\top \nabla_\theta \mathcal{L}_{in}(\theta_\tau, \beta_{\tau+1}) \geq C ||\nabla_\theta \mathcal{L}_{in}(\theta_\tau, \beta_{\tau+1})||^2 \tag{8}$$

*holds for a constant $C \geq 0$ in every step $\tau$,[2] and the learning rate satisfies $\mu \leq \frac{2C}{L}$, then the outer loss decreases along with each iteration: $\mathcal{L}_{out}(\theta_{\tau+1}) \leq \mathcal{L}_{out}(\theta_\tau)$, and the equality holds if $\nabla_\beta \mathcal{L}_{out}(\theta_\tau) = 0$ or $\theta_{\tau+1} = \theta_\tau$.*

**Remark 1.** *The inequality in the assumption of Theorem 1 (Eqn. 8) indicates that the directions of the gradients of $\mathcal{L}_{out}$ and $\mathcal{L}_{in}$ with respect to $\theta$ should be close. Intuitively only when the two gradient directions align, we can decrease the evaluation loss $\mathcal{L}_{out}$ by updating $\theta$ with $\mathcal{L}_{in}$.*

Theorem 1 ensures that the confidence and the imitation learning parameters monotonically decrease the outer loss. When the gradient of the outer loss with respect to $\beta$ is zero, $\beta$ converges to the optimal confidence that minimizes the outer loss, *i.e.,* , $\beta^*$ in Eqn. (1). With the optimal confidence, we can learn a well-performing policy from more useful demonstrations by reweighting them. Thus, the learned imitation model induces lower outer loss (has higher-quality) than the imitation learning model learned from the original demonstrations in the dataset without reweighting.

---

[2] We remove $(s, a)$ in $\mathcal{L}_{\text{in}}$ for notation simplicity.

**Theorem 2.** *(Convergence Rate) Under the assumptions in Theorem 1, let*

$$g(\theta, \beta) = \theta - \mu \nabla_\theta \mathcal{L}_{in}(s, a; \theta, \beta) \tag{9}$$

*We assume that $\mathcal{L}_{out}(g(\theta, \beta))$ is Lipschitz-smooth w.r.t. $\beta$ with constant $L_1$, $\mathcal{L}_{in}$ and $\mathcal{L}_{out}$ have $\sigma$-bounded gradients, and the norm of $\nabla_\beta \nabla_\theta \mathcal{L}_{in}(\theta; \beta)$ is bounded by $\sigma_1$. $L$ is the Lipschitz-smooth constant for $\mathcal{L}_{out}$ w.r.t. $g(\theta, \beta)$ as shown in Theorem 1. Consider the total training steps as $T$, we set $\alpha = \frac{C_1}{\sqrt{T}}$, for some constant $C_1$ where $0 < C_1 \leq \frac{2}{L_1}$ and $\mu = \frac{C_2}{T}$ for some constant $C_2$. Then:*

$$\min_{1 \leq \tau \leq T} \mathbb{E}[||\nabla_\beta \mathcal{L}_{out}(\theta_\tau)||^2] \leq O\left(\frac{1}{\sqrt{T}}\right). \tag{10}$$

**Remark 2.** *The assumptions of Theorem 2 are Lipschiz-smoothness and bounded first-order and second-order gradients of $\mathcal{L}_{in}$ and $\mathcal{L}_{out}$, which are satisfied for typical $\mathcal{L}_{in}$ and $\mathcal{L}_{out}$ such as the cross-entropy loss of AIRL and the ranking loss in our implementation of CAIL in Section 4.3.*

With the bound on the convergence rate, the gradient of the outer loss with respect to $\beta$ is gradually getting close to 0, which means that $\beta$ gradually converges to the optimal confidence $\beta^*$ that minimizes the outer loss if $\mathcal{L}_{\text{out}}$ is convex with respect to $\beta$.

## 4.3 An Implementation of CAIL

To implement CAIL, we need to adopt an imitation learning algorithm whose imitation loss will be the inner loss. We also need to design an outer loss on the imitation learning algorithm to evaluate the quality of imitation given some evaluation data $\mathcal{D}_E$ (*e.g.* partial ranking annotations).

Based on the above considerations, as an instance of the implementation of CAIL, we use Adversarial Inverse Reinforcement Learning (AIRL) [14] as our imitation learning model. We use the imitation loss of AIRL as the inner loss, and a ranking loss (based on a partial ranking of trajectories) as the outer loss. AIRL and the ranking loss are compatible since AIRL can induce the reward function from the discriminator within the model, and the ranking loss can penalize the mismatches of the trajectory rankings computed by the induced reward function and the ground-truth rankings from the evaluation data $\mathcal{D}_E$. Furthermore, the implementation only requires the ranking of a subset of demonstrations $\{\xi_i\}_{i=1}^m \subset \Xi$, *i.e.*, , $\mathcal{D}_E = \eta_{\xi_1} \geq \eta_{\xi_2} \geq \cdots \geq \eta_{\xi_m}$, which is much easier to access than the exact confidence value annotations [6, 19] since confidence not only reflects the rankings of different demonstrations but also how much one demonstration is better than the other.

AIRL consists of a generator $G$ parameterized by $\theta_G$ as the policy, and a discriminator parameterized by $\theta_D$. The generator and the discriminator are trained in an adversarial manner as in [15] to match the occupancy measures of the policy and the demonstrations. We write the loss $\mathcal{L}_{\text{in}}$ as:

$$\mathcal{L}_{\text{in}}^D(s, a; \theta^D, \beta) = \mathbb{E}_{(s,a) \sim \beta(s,a)p_{\pi d(s,a)}}[-\log D(s, a)] + \mathbb{E}_{(s,a) \sim \pi_{\theta G}}[-\log(1 - D(s, a))], \tag{11}$$

$$\mathcal{L}_{\text{in}}^G(s, a; \theta^G) = \mathbb{E}_{(s,a) \sim \pi_{\theta G}}[\log D(s, a) - \log(1 - D(s, a))], \tag{12}$$

where $\mathcal{L}_{\text{in}}^D$ is the inner loss for the discriminator, $\mathcal{L}_{\text{in}}^G$ is the inner loss for the generator and $\pi_{\theta G}$ is the policy derived from the generator. The discriminator $D$ is learned by minimizing the loss $\mathcal{L}_{\text{in}}^D$, which aims to discriminates the state-action pair $(s, a)$ drawn from $\pi_{\theta G}$ and the state-action pair $(s, a)$ drawn from $\pi^d$. The generator parameter $\theta^G$ is trained to minimize the loss $\mathcal{L}_{\text{in}}^G$, which enables the generator to generate state-action pairs that are similar to the state transitions in the demonstrations.

For the outer loss, AIRL approximates the reward function by the discriminator parameters, *i.e.*, $\mathcal{R}'_{\theta D}$. We compute $\eta'_{\xi_i} = \sum_{t=0}^N \gamma^t \mathcal{R}'_{\theta D}(s_t, a_t)$ as the expected return of a trajectory using the reward $\mathcal{R}'_{\theta D}$. Then we penalize the mismatches of the rankings derived by $\eta'_{\xi_i}$ and the ground-truth rankings:

$$\mathcal{L}_{\text{out}}(\theta_D) = \sum_i \sum_{j>i} \text{RK}\left[\eta'_{\xi_i}, \eta'_{\xi_j}; \mathbb{I}[\eta_{\xi_i} > \eta_{\xi_j}]\right], \tag{13}$$

where $\mathbb{I}[\eta_{\xi_i} > \eta_{\xi_j}]$ is 1 if $\eta_{\xi_i} > \eta_{\xi_j}$ and otherwise is $-1$. RK is defined as a revised version of the widely-used margin ranking loss with margin as 0:

$$\text{RK}\left[\eta'_{\xi_i}; \eta'_{\xi_j}, \eta_{\xi_i}, \eta_{\xi_j}\right] = \begin{cases} \max(0, -\mathbb{I}[\eta_{\xi_i} > \eta_{\xi_j}](\eta'_{\xi_i} - \eta'_{\xi_j})), & |\eta'_{\xi_i} - \eta'_{\xi_j}| > \epsilon \\ \max(0, \frac{1}{4\epsilon}(\mathbb{I}[\eta_{\xi_i} > \eta_{\xi_j}](\eta'_{\xi_i} - \eta'_{\xi_j}) - \epsilon)^2), & |\eta'_{\xi_i} - \eta'_{\xi_j}| \leq \epsilon \end{cases} \tag{14}$$

We revised the original margin ranking loss within a $\epsilon$ range around the point of $(\eta'(\xi_i) - \eta'(\xi_j)) = 0$ to make it Lipschitz smooth. If we adopt small enough $\epsilon$, the functionality of the revised marginal ranking loss is close to the original one. In all the experiments, we use $\epsilon = 10^{-5}$.

## 5 Experiments

In this section, we conduct experiments on the implementation of CAIL in Sec. 4.3. We verify the efficacy of the CAIL in simulated and real-world environments. We report the results on various compositions of demonstrations with varying optimality. **The code is available on our website**[3]

We conduct experiments in four environments including two MuJoCo environments (Reacher and Ant) [28] in OpenAI Gym [7], one Franka Panda Arm[4] simulation environment, and one real robot environment with a UR5e robot arm[5]. For each environment, we collect a mixture of optimal and non-optimal demonstrations with different optimality to show the efficacy of CAIL. We investigate the performance with respect to the optimality of demonstrations ranging from failures to near-optimal or optimal demonstrations. We provide the implementation details and more results on the sensitivity of the parameters, and visualize the learned confidence in the supplementary materials.

**Source of Demonstrations.** For MuJoCo environments, following the demonstration collecting method in [30], we train a reinforcement learning algorithm and select four intermediate policies as policies with varying optimality and the converged policy as the optimal policy, so that the demonstrations range from worse-than-random ones to near-optimal ones. We draw $20\%$ of demonstrations from each policy. For the RL algorithm, we use SAC [16] for the Reacher environment and PPO [24] for the Ant environment. For the Franka Panda Arm simulation and the real robot environment with UR5e, we hand-craft demonstrations with optimality varying continuously from near-optimal ones to unsuccessful ones to approximate the demonstration collecting process from demonstrators with different levels of expertise. We label only $5\%$ of the demonstrated trajectories with rankings since we target realistic settings where only a small number of rankings are available for the demonstrations.

**Baselines.** We compare CAIL with the most relevant works in our problem setting including: the state-of-the-art standard imitation learning algorithms: GAIL [18], AIRL [14], imitation learning from suboptimal demonstration methods including two confidence-based methods, 2IWIL and IC-GAIL [30], and three ranking-based methods, T-REX [8], D-REX [9], and SSRR [12]. GAIL and AIRL learn directly from the mixture of optimal and non-optimal demonstrations. T-REX needs demonstrations paired with rankings, so we provide the same number of rankings as our approach. For D-REX and SSRR, we further generate rankings by disturbing demonstrations as done in their papers. For 2IWIL and IC-GAIL—that need a subset of demonstrations labeled with confidence—we label the subset of ranked demonstrations with evenly-spaced confidence, *i.e.,* , the highest expected return as confidence $1$, and the lowest expected return as $0$. This is a reasonable approximation of the confidence score with no prior knowledge available. For a fair comparison, we re-implement 2IWIL with AIRL as its backbone imitation learning method. For the RL algorithm in T-REX, D-REX, and SSRR, we also use PPO. DPS [19] requires interactively collecting demonstrations and the approach in Cao *et al.* [11] requires the ground truth reward of demonstrations, which are both not implementable under the assumptions in our setting, so we do not include them.

### 5.1 Results

**Reacher and Ant.** In the Reacher, the end effector of the arm is supposed to reach a final location. Figure 2(a) shows the optimal trajectories of the joint and the end effector in green, which illustrates the policy reaching the location with the minimum energy cost, and the trajectories with lower optimality in red and orange, where the agent just spins around the center and wastes energy without reaching the target. We collect 200 trajectories in total, where each trajectory has $50$ interaction steps.

In Ant, the agent has four legs, each with two links and two joints. Its goal is to move in the x-axis direction as fast as possible. Figure 2(b) illustrates the demonstrated trajectories, where green shows the optimal one, and red shows suboptimal trajectories (darker colors show lower optimality). In

---

[3]https://sites.google.com/view/cail

[4]https://www.franka.de

[5]https://www.universal-robots.com/products/ur5-robot

optimal demonstrations, the agent moves quickly along the x-axis, while in suboptimal ones, it moves slowly to other directions. We collect trajectories with 200,000 interaction steps in total.

As shown in Figure 2(e) and 2(f), CAIL achieves the highest expected return compared to the other methods and experiences fast convergence. For Reacher, the p-value[6] between CAIL and the closest baseline method, T-REX, is $5.4054 \times 10^{-6}$ (statistically significant). For Ant, the p-value between CAIL and the closest baseline method, 2IWIL, is $0.1405$. CAIL outperforms standard imitation learning methods, GAIL and AIRL, because CAIL selects more useful demonstrations, and avoids the negative influence of harmful demonstrations. We observe that 2IWIL and IC-GAIL do not perform well because neighboring demonstrations in a given ranking are not guaranteed to have the same distance in terms of confidence score and thus the evenly-spaced confidence values derived from rankings are likely not accurate. All the ranking-based methods do not perform well. For T-REX, the potential reason can be that the rankings of a subset of demonstrations are not enough to learn a generalizable reward function covering states. For D-REX and SSRR, the automatically generated rankings can be incorrect since we also have unsuccessful demonstrations— which can at times be worse than random actions—and perturbing such demonstrations is not guaranteed to produce demonstrations that imply rankings.

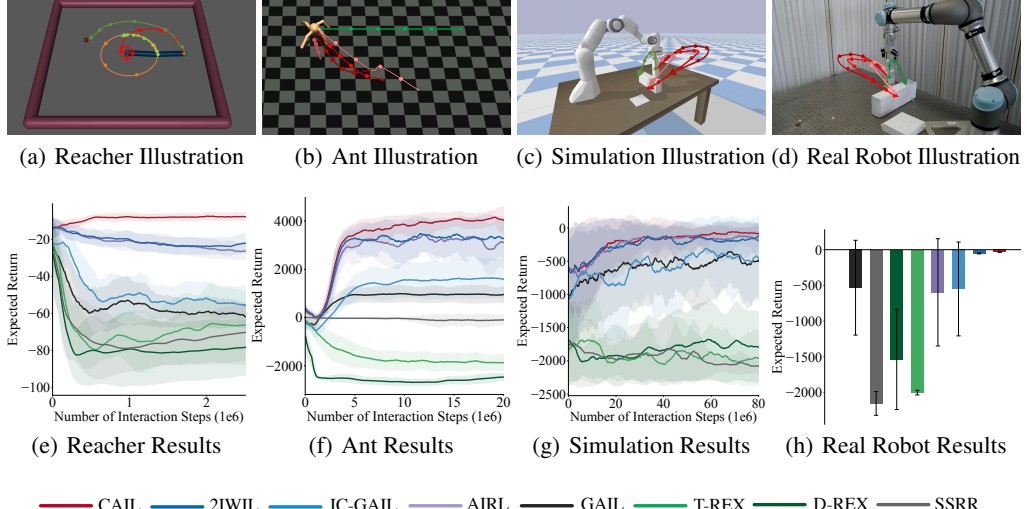

(a) Reacher Illustration    (b) Ant Illustration    (c) Simulation Illustration    (d) Real Robot Illustration

(e) Reacher Results    (f) Ant Results    (g) Simulation Results    (h) Real Robot Results

CAIL — 2IWIL — IC-GAIL — AIRL — GAIL — T-REX — D-REX — SSRR

Figure 2: (a) Reacher, (b) Ant, (c) Simulated Panda Robot Arm, (d) Real UR5e Robot Arm. In (a-d), the green trajectories indicate the optimal demonstrations, while the red and orange trajectories indicate demonstrations with varying optimality. (e-g) The expected return with respect to the number of interaction steps. (h) The expected return of the converged policies for UR5e Robot Arm.

**Robot Arm.** We further conduct experiments in more realistic environments: a simulated Franka Panda Arm and a real UR5e robot arm. As shown in Figure 2(c) and 2(d), we design a task to let the robot arm pick up a bottle, avoid the obstacle, and put the bottle on a target platform. In the optimal demonstrations in green, the arm takes the shortest path to avoid the obstacle, and puts the bottle on the target, while in suboptimal ones in red (where, similar to before, the brightness of the trajectories indicates their optimality), the arm detours, does not reach the target, and even at times collides with the obstacle. The suboptimal demonstrations represent a wide range of optimality from near-optimal ones (small detour) to adversarial ones (colliding). We vary the initial position of the robot end-effector and the goal position within an initial area and goal area respectively. For both simulated and real robot environments, we collect trajectories with 200,000 interaction steps in total.

As shown in Figure 2(g) and 2(h), CAIL outperforms other methods in expected return in both the simulated and real robot environments. For the simulated robot arm environment, the p-value between CAIL and the closest baseline, 2IWIL, is $0.0974$. For the real robot environment, the p-value between CAIL and the closest baseline, AIRL, is $0.0209$ (statistically significant). In particular, in the real robot environment, CAIL achieves a low standard deviation while other methods especially AIRL, IC-GAIL, D-REX and GAIL suffer from an unstable performance. The results demonstrate that

---

[6]All p-values are computed by the student's t-test and the null hypothesis is the performance of CAIL is equal to or smaller than the baseline methods.

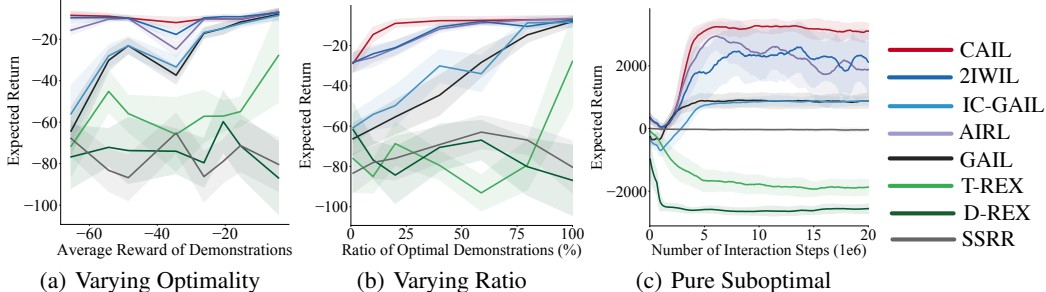

|  | (a) Varying Optimality | (b) Varying Ratio | (c) Pure Suboptimal |

Figure 3: (a-b) The Expected Return with respect to different optimality of demonstrations in the Reacher environment, where the different optimality are created by varying the optimality of non-optimal demonstrations, and varying the ratio of optimal demonstrations. (c) Results for learning from only non-optimal demonstrations in the Ant environment.

CAIL can work stably in the real robot environment. We report the success rate—rate that the robot successfully reaches the target within the time limit without colliding with the obstacle—and videos of sample policy rollouts in the supplementary materials.

**Demonstrations with Different Optimality.** We show the performance of different methods with demonstrations at different levels of optimality in the Reacher environment. We fix $20\%$ of the total demonstrations to be optimal and make the remaining $80\%$ demonstration drawn from the same suboptimal policy. We vary the optimality of this policy to investigate the performance change with respect to different optimalities. Another way to obtain different optimality is to vary the ratio of optimal demonstrations. We show the results of both varying optimality in Figure 3(a) and 3(b) respectively. We observe that CAIL consistently outperforms or performs comparably to other methods with demonstrations at different optimality. Also, CAIL performs more stably while the baselines suffer from a performance drop at specific optimality levels.

**Learning from Only Non-optimal Demonstrations.** We verify that CAIL can also learn from solely non-optimal demonstrations without relying on any optimal demonstrations. We remove the optimal demonstrations in the Ant environment and use the remaining demonstrations to conduct imitation learning. As shown in Figure 3(c), CAIL still achieves the best performance among all the methods, which demonstrates that even with all demonstrations being non-optimal, CAIL still can learn useful knowledge from those demonstrations with higher expected return and induce a better policy. The highest p-value between CAIL and the closest baseline (2IWIL) is $0.0067$, which indicates the performance gain is statistically significant. We observe that the performance of AIRL first increases and then decreases. This is because even though the demonstrations are suboptimal, there are potentially optimal state-action transitions leading to the initial high performance. However, at this early training stage, the AIRL model does not converge yet and the model parameters can still change rapidly. After training a sufficient number of steps, the AIRL model observes both useful and less useful transitions and converges to the average return of all the demonstrations.

## 6 Conclusion

**Summary.** We propose a general learning framework, Confidence-Aware Imitation Learning, for imitation learning from demonstrations with varying optimality. We adopt standard imitation learning algorithms with their corresponding imitation loss (inner loss), and leverage an outer loss to evaluate the quality of the imitation learning model. We simultaneously learn a confidence score over the demonstrations using the outer loss and learn the policy through optimizing the inner loss over the confidence-reweighted distribution of demonstrations. Our framework is applicable to any imitation learning model with compatible choices of inner and outer losses. We provide theoretical guarantees on the convergence of CAIL and show that the learned policy outperforms baselines on various simulated and real-world environments under demonstrations with varying optimality.

**Limitations and Future Work.** Although we propose a flexible framework to address the problem of imitation learning from demonstrations with varying optimality, our work is limited in a few ways: To learn a well-performing policy, we still require that the dataset consists of demonstrations that encode useful knowledge for policy learning. We also require that the demonstrations and the imitation agent have the same dynamics. In the future, we plan to learn from demonstrations with more failures and relax the assumptions of the demonstrator and imitator having the same dynamics.

# 7 Acknowledgement

We would like to thank Tong Xiao for inspiring discussions about the backbone imitation learning algorithms and her help on the experiments. This work is funded by Tsinghua GuoQiang Research Institute, FLI grant RFP2-000, NSF Awards 1941722 and 1849952, and DARPA HiCoN project.

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
