# Appendix of Confidence-Aware Imitation Learning from Demonstrations with Varying Optimality

## A  Proofs

In this section, we provide the proofs of the theorems in Section 4.2 of the main text.

### A.1  Preliminaries

**Definition 1.** *(Lipschitz-smooth) Function $f(x) : \mathbb{R}^d \to \mathbb{R}$ is Lipschitz-smooth with constant $L$ if*

$$||\nabla f(x) - \nabla f(y)|| \le L||x - y||, \quad \forall x, y \in \mathbb{R}^d \tag{1}$$

**Lemma 1.** *If function $f(x)$ is Lipschitz-smooth with constant $L$, then the following inequality holds:*

$$(\nabla f(x) - \nabla f(y))^T (x - y) \le L||x - y||^2 \tag{2}$$

*Proof.* The proof is straight forward that

$$
\begin{aligned}
&(\nabla f(x) - \nabla f(y))^T (x - y) \\
&\le ||\nabla f(x) - \nabla f(y)|| \cdot ||x - y|| \\
&\le L||x - y||^2
\end{aligned}
\tag{3}
$$

The first equation follows from the Cauchy-Schwarz inequality, and the second inequality comes from the definition of Lipschitz-smooth. $\square$

**Lemma 2.** *If function $f(x)$ is Lipschitz-smooth with constant $L$, then the following inequality holds:*

$$f(y) \le f(x) + \nabla f(x)^T(y - x) + \frac{L}{2}||y - x||^2, \quad \forall x, y \tag{4}$$

*Proof.* Define $g(t) = f(x + t(y - x))$. If $f(x)$ is Lipschitz-smooth with constant $L$, then from Lemma 1, we have

$$
\begin{aligned}
&g'(t) - g'(0) \\
&= (\nabla f(x + t(y - x)) - \nabla f(x))^T (y - x) \\
&= \frac{1}{t} (\nabla f(x + t(y - x)) - \nabla f(x))^T ((x + t(y - x)) - x) \\
&\le \frac{L}{t} ||t(y - x)||^2 = tL||y - x||^2
\end{aligned}
\tag{5}
$$

35th Conference on Neural Information Processing Systems (NeurIPS 2021).

We then integrate this equation from $t = 0$ to $t = 1$:

$$
\begin{aligned}
f(y) = g(1) &= g(0) + \int_0^1 g'(t)dt \\
&\leq g(0) + \int_0^1 g'(0)dt + \int_0^1 tL||y - x||^2 dt \\
&= g(0) + g'(0) + \frac{L}{2}||y - x||^2 \\
&= f(x) + \nabla f(x)^T(y - x) + \frac{L}{2}||y - x||^2
\end{aligned}
\tag{6}
$$

$\square$

## A.2 Proof of Theoretical Results

In this section, we provide the proofs of the theorems proposed in this paper. First, we provide the proof of Theorem 1 in the main text.

**Theorem 1.** *(Convergence) Suppose the outer loss $\mathcal{L}_{out}$ is Lipschitz-smooth with constant L, the inequality*

$$
\nabla_\theta \mathcal{L}_{out}(\theta_{\tau+1})^\top \nabla_\theta \mathcal{L}_{in}(\theta_\tau, \beta_{\tau+1}) \geq C||\nabla_\theta \mathcal{L}_{in}(\theta_\tau, \beta_{\tau+1})||^2
\tag{7}
$$

*holds for a constant $C \geq 0$ in every step $\tau$[1], and the learning rate satisfies $\mu \leq \frac{2C}{L}$, then the outer loss decreases along with each iteration: $\mathcal{L}_{out}(\theta_{\tau+1}) \leq \mathcal{L}_{out}(\theta_\tau)$, and the equality holds if $\nabla_\beta \mathcal{L}_{out}(\theta_\tau) = 0$ or $\theta_{\tau+1} = \theta_\tau$.*

*Proof.* Since $\mathcal{L}_{out}$ is Lipschitz-smooth, following Lemma 2, we have

$$
\begin{aligned}
&\mathcal{L}_{out}(\theta_{\tau+1}) - \mathcal{L}_{out}(\theta_\tau) \\
&\leq \nabla_\theta \mathcal{L}_{out}(\theta_\tau)^T(\theta_{\tau+1} - \theta_\tau) + \frac{L}{2}||(\theta_{\tau+1} - \theta_\tau)||^2 \\
&= -\mu \nabla_\theta \mathcal{L}_{out}(\theta_{\tau+1})^T \nabla_\theta \mathcal{L}_{in}(\theta_\tau, \beta_{\tau+1}) \\
&\quad + \frac{L}{2}\mu^2||\nabla_\theta \mathcal{L}_{in}(\theta_\tau, \beta_{\tau+1})||^2 \\
&\leq -\left(\mu C - \frac{L}{2}\mu^2\right)||\nabla_\theta \mathcal{L}_{in}(\theta_\tau, \beta_{\tau+1})||^2 \\
&\leq 0
\end{aligned}
\tag{8}
$$

The first inequality comes from Lemma 2, and the second inequality holds because we update $\theta_\tau$ to $\theta_{\tau+1}$ only when $\nabla_\theta \mathcal{L}_{out}(\theta_{\tau+1})^T \nabla_\theta \mathcal{L}_{in}(\theta_\tau, \beta_{\tau+1}) \geq C||\nabla_\theta \mathcal{L}_{in}(\theta_\tau, \beta_{\tau+1})||^2$ holds, otherwise $\theta_{\tau+1} = \theta_\tau$ so $\mathcal{L}_{out}(\theta_{\tau+1}) = \mathcal{L}_{out}(\theta_\tau)$. The third inequality holds because we choose the learning rate to satisfy $\mu \leq \frac{2C}{L}$.

Then if $\nabla_\theta \mathcal{L}_{out}(\theta_\tau) = 0$, and if Eqn. (7) is satisfied, we have $\nabla_\theta \mathcal{L}_{in}(\theta_\tau, \beta_{\tau+1}) = 0$. Following the updating rule of $\alpha$ in Eqn. (10), we can derive $\theta_{\tau+1} = \theta_\tau$, so $\mathcal{L}_{out}(\theta_{\tau+1}) = \mathcal{L}_{out}(\theta_\tau)$. Besides, if Eqn. (11) is not satisfied, we also have $\theta_{\tau+1} = \theta_\tau$, and thus $\mathcal{L}_{out}(\theta_{\tau+1}) = \mathcal{L}_{out}(\theta_\tau)$. $\square$

We now provide the proof of Theorem 2 on convergence rate of the algorithm.

**Theorem 2.** *(Convergence Rate) Under the assumptions in Theorem 1, let*

$$
g(\theta, \beta) = \theta - \mu \nabla_\theta \mathcal{L}_{in}(s, a; \theta, \beta).
\tag{9}
$$

*We assume that $\mathcal{L}_{out}(g(\theta, \beta))$ is Lipschitz-smooth w.r.t. $\beta$ with constant $L_1$, $\mathcal{L}_{in}$ and $\mathcal{L}_{out}$ have $\sigma$-bounded gradients, and the norm of $\nabla_\beta \nabla_\theta \mathcal{L}_{in}(\theta; \beta)$ is bounded by $\sigma_1$. $L$ is the Lipschitz-smooth constant for $\mathcal{L}_{out}$ w.r.t. $g(\theta, \beta)$ as shown in Theorem 1. Consider the total training steps as $T$, we set $\alpha = \frac{C_1}{\sqrt{T}}$, for some constant $C_1$ where $0 < C_1 \leq \frac{2}{L_1}$ and $\mu = \frac{C_2}{T}$ for some constant $C_2$. CAIL can achieve:*

$$
\min_{1 \leq \tau \leq T} \mathbb{E}[||\nabla_\beta \mathcal{L}_{out}(\theta_\tau)||^2] \leq O\left(\frac{1}{\sqrt{T}}\right).
\tag{10}
$$

---

[1]We remove $(s, a)$ in $\mathcal{L}_{in}$ for notation simplicity.

*Proof.* According to the update rule of $\theta$, we have

$$
\begin{aligned}
&\mathcal{L}_{\text{out}}(\theta_{\tau+1}) - \mathcal{L}_{\text{out}}(\theta_\tau) \\
&= \mathcal{L}_{\text{out}}(\theta_\tau - \mu \nabla_\theta \mathcal{L}_{\text{in}}(\theta_\tau, \beta_{\tau+1})) \\
&\quad - \mathcal{L}_{\text{out}}(\theta_{\tau-1} - \mu \nabla_\theta \mathcal{L}_{\text{in}}(\theta_{\tau-1}, \beta_\tau)) \\
&= \{\mathcal{L}_{\text{out}}(\theta_\tau - \mu \nabla_\theta \mathcal{L}_{\text{in}}(\theta_\tau, \beta_{\tau+1})) \\
&\quad - \mathcal{L}_{\text{out}}(\theta_{\tau-1} - \mu \nabla_\theta \mathcal{L}_{\text{in}}(\theta_{\tau-1}, \beta_{\tau+1}))\} \\
&\quad + \{\mathcal{L}_{\text{out}}(\theta_{\tau-1} - \mu \nabla_\theta \mathcal{L}_{\text{in}}(\theta_{\tau-1}, \beta_{\tau+1})) \\
&\quad - \mathcal{L}_{\text{out}}(\theta_{\tau-1} - \mu \nabla_\theta \mathcal{L}_{\text{in}}(\theta_{\tau-1}, \beta_\tau))\} \\
&= \{\mathcal{L}_{\text{out}}(g(\theta_\tau, \beta_{\tau+1})) - \mathcal{L}_{\text{out}}(g(\theta_{\tau-1}, \beta_{\tau+1}))\} \\
&\quad + \{\mathcal{L}_{\text{out}}(g(\theta_{\tau-1}, \beta_{\tau+1})) - \mathcal{L}_{\text{out}}(g(\theta_{\tau-1}, \beta_\tau))\}
\end{aligned}
\tag{11}
$$

We remove $(s, a)$ in the $\mathcal{L}_{\text{in}}$ for notation convenience. For the first term,

$$
\begin{aligned}
&\mathcal{L}_{\text{out}}(g(\theta_\tau, \beta_{\tau+1})) - \mathcal{L}_{\text{out}}(g(\theta_{\tau-1}, \beta_{\tau+1})) \\
&\leq \nabla \mathcal{L}_{\text{out}}(g(\theta_{\tau-1}, \beta_{\tau+1}))^T \Delta g + \frac{L}{2} ||\Delta g||^2
\end{aligned}
\tag{12}
$$

where

$$
\begin{aligned}
\Delta g &= g(\theta_\tau, \beta_{\tau+1}) - g(\theta_{\tau-1}, \beta_{\tau+1}) \\
&= [\theta_\tau - \mu \nabla_\theta \mathcal{L}_{\text{in}}(\theta_\tau, \beta_{\tau+1})] \\
&\quad - [\theta_{\tau-1} - \mu \nabla_\theta \mathcal{L}_{\text{in}}(\theta_{\tau-1}, \beta_{\tau+1})] \\
&= -\mu [\nabla_\theta \mathcal{L}_{\text{in}}(\theta_\tau, \beta_{\tau+1}) + \nabla_\theta \mathcal{L}_{\text{in}}(\theta_{\tau-1}, \beta_\tau) \\
&\quad - \nabla_\theta \mathcal{L}_{\text{in}}(\theta_{\tau-1}, \beta_{\tau+1})]
\end{aligned}
\tag{13}
$$

Since $\mathcal{L}_{\text{in}}$ has $\sigma$-bounded gradients, we take the norm on both sides, and use the triangle inequality, so

$$
||\Delta g|| \leq 3\mu\sigma
\tag{14}
$$

Substitute this into Eqn. (12), we have

$$
\begin{aligned}
&\mathcal{L}_{\text{out}}(g(\theta_\tau, \beta_{\tau+1})) - \mathcal{L}_{\text{out}}(g(\theta_{\tau-1}, \beta_{\tau+1})) \\
&\leq 3\mu\sigma^2 + \frac{9}{2} L \mu^2 \sigma^2
\end{aligned}
\tag{15}
$$

And for the second term,

$$
\begin{aligned}
&\mathcal{L}_{\text{out}}(g(\theta_{\tau-1}, \beta_{\tau+1})) - \mathcal{L}_{\text{out}}(g(\theta_{\tau-1}, \beta_\tau)) \\
&\leq \nabla_\beta \mathcal{L}_{\text{out}}(g(\theta_{\tau-1}, \beta_\tau))^T (\beta_{\tau+1} - \beta_\tau) + \frac{L_1}{2} ||\beta_{\tau+1} - \beta_\tau||^2 \\
&= -\alpha \nabla_\beta \mathcal{L}_{\text{out}}(g(\theta_{\tau-1}, \beta_\tau))^T \nabla_\beta \mathcal{L}_{\text{out}}(g(\theta_\tau, \beta_\tau)) \\
&\quad + \frac{L_1 \alpha^2}{2} ||\nabla_\beta \mathcal{L}_{\text{out}}(g(\theta_\tau, \beta_\tau))||^2 \\
&= -(\alpha - \frac{L_1 \alpha^2}{2}) ||\nabla_\beta \mathcal{L}_{\text{out}}(g(\theta_\tau, \beta_\tau))||^2 \\
&\quad + \alpha (\nabla_\beta \mathcal{L}_{\text{out}}(g(\theta_\tau, \beta_\tau)) - \nabla_\beta \mathcal{L}_{\text{out}}(g(\theta_{\tau-1}, \beta_\tau)))^T \nabla_\beta \mathcal{L}_{\text{out}}(g(\theta_\tau, \beta_\tau))
\end{aligned}
\tag{16}
$$

Since $\nabla_\beta \nabla_\theta \mathcal{L}_{\text{in}}(\theta, \beta)$ is bounded by $\sigma_1$ and $\mathcal{L}$ has $\sigma$-bounded gradients, then

$$
\begin{aligned}
&\nabla_\beta \mathcal{L}_{\text{out}}(g(\theta, \beta)) \\
&= \nabla_\beta g(\theta, \beta)^T \nabla_g \mathcal{L}_{\text{out}}(g(\theta, \beta)) \\
&= -\mu \nabla_\beta \nabla_\theta \mathcal{L}_{\text{in}}(\theta, \beta)^T \nabla_g \mathcal{L}_{\text{out}}(g(\theta, \beta)) \\
&\leq \mu \sigma \sigma_1
\end{aligned}
\tag{17}
$$

So

$$
\begin{aligned}
&\mathcal{L}_{\text{out}}(g(\theta_{\tau-1}, \beta_{\tau+1})) - \mathcal{L}_{\text{out}}(g(\theta_{\tau-1}, \beta_\tau)) \\
&\leq -(\alpha - \frac{L_1 \alpha^2}{2}) ||\nabla_\beta \mathcal{L}_{\text{out}}(g(\theta_\tau, \beta_\tau))||^2 \\
&\quad + 2\alpha \mu \sigma \sigma_1
\end{aligned}
\tag{18}
$$

Combining the two parts, we can derive that

$$
\mathcal{L}_{\text{out}}(\theta_{\tau+1}) - \mathcal{L}_{\text{out}}(\theta_\tau)
$$
$$
\leq -(\alpha - \frac{L_1\alpha^2}{2})||\nabla_\beta \mathcal{L}_{\text{out}}(g(\theta_\tau, \beta_\tau))||^2 \tag{19}
$$
$$
+ 3\mu\sigma^2 + \frac{9}{2}L\mu^2\sigma^2 + 2\alpha\mu\sigma\sigma_1
$$

Summing up both sides from $\tau = 1$ to $T$, and rearranging the terms, we can derive that

$$
\sum_{\tau=1}^{T}(\alpha - \frac{L_1\alpha^2}{2})||\nabla_\beta \mathcal{L}_{\text{out}}(\theta_\tau)||^2
$$
$$
\leq \mathcal{L}_{\text{out}}(\theta_1) - \mathcal{L}_{\text{out}}(\theta_{T+1}) + T\left(3\mu\sigma^2 + \frac{9}{2}L\mu^2\sigma^2 + 2\alpha\mu\sigma\sigma_1\right) \tag{20}
$$

Since $\alpha - \frac{L_1\alpha^2}{2} \geq 0$, we have

$$
\min_\tau \mathbb{E}[||\nabla_\beta \mathcal{L}_{\text{out}}(\theta_t)||^2]
$$
$$
\leq \frac{\sum_{\tau=1}^{T}(\alpha - \frac{L_1\alpha^2}{2})||\nabla_\beta \mathcal{L}_{\text{out}}(\theta_t)||^2}{T(\alpha - \frac{L_1\alpha^2}{2})}
$$
$$
\leq \frac{1}{T\alpha(1 - \frac{L_1\alpha}{2})}\left[\mathcal{L}_{\text{out}}(\theta_1) - \mathcal{L}_{\text{out}}(\theta_{T+1})\right.
$$
$$
\left. +T\left(3\mu\sigma^2 + \frac{9}{2}L\mu^2\sigma^2 + 2\alpha\mu\sigma\sigma_1\right)\right]
$$
$$
\leq \frac{1}{\alpha\sqrt{T}(\sqrt{T}-1)}\left[\mathcal{L}_{\text{out}}(\theta_1) - \mathcal{L}_{\text{out}}(\theta_{T+1})\right. \tag{21}
$$
$$
\left. +T\left(3\mu\sigma^2 + \frac{9}{2}L\mu^2\sigma^2 + 2\alpha\mu\sigma\sigma_1\right)\right]
$$
$$
= \frac{\mathcal{L}_{\text{out}}(\theta_1) - \mathcal{L}_{\text{out}}(\theta_{T+1})}{\alpha\sqrt{T}(\sqrt{T}-1)} + \frac{\sigma\mu\sqrt{T}}{\alpha(\sqrt{T}-1)}\left(3\sigma + \frac{9}{2}L\mu\sigma + 2\alpha\sigma_1\right)
$$
$$
= \frac{\mathcal{L}_{\text{out}}(\theta_1) - \mathcal{L}_{\text{out}}(\theta_{T+1})}{C_1(\sqrt{T}-1)} + \frac{\sigma C_2}{C_1(\sqrt{T}-1)}\left(3\sigma + \frac{9}{2}L\mu\sigma + 2\alpha\sigma_1\right)
$$
$$
= O\left(\frac{1}{\sqrt{T}}\right)
$$

The second inequality holds since $1 - \frac{L_1\alpha}{2} \geq 1 - \frac{1}{\sqrt{T}}$. $\hfill\square$

**Theorem 3.** *RK in Eqn. (14) in the main text is Lipschitz.*

*Proof.* We consider the case where $\eta_{\xi_i} > \eta_{\xi_j}$. The case for $\eta_{\xi_i} <= \eta_{\xi_j}$ can be demonstrated similarly. We prove that RK is Lipschitz by definition. Let $z = \xi_i' - \xi_j'$, the derivative of RK is

$$
\nabla RK_z = \begin{cases} 0 & z > \epsilon, \\ \frac{1}{2\epsilon}(z - \epsilon) & -\epsilon \leq z \leq \epsilon, \\ -1 & z < -\epsilon \end{cases}
$$

Thus, the second-order derivative of $RK$ satisfies that $|\nabla(\nabla RK_z)_z| \leq \frac{1}{2\epsilon}$. If we take $L > \frac{1}{2\epsilon}$, then $\forall z_1, z_2, |\nabla(z_1) - \nabla(z_2)| < L|z_1 - z_2|$. This proves that RK is Lipschitz smooth.

$\hfill\square$

# B Experiments

In this section, we provide additional experimental details and results.

## B.1 Experimental Details

We first go over the implementation details. We will then describe the environments and details of running experiments including train/validation/test splits, the number of evaluation runs, the average time for each run, and the used computing infrastructure.

### B.1.1 Implementation Details

We implement CAIL based on a PPO-based AIRL. The actor and the critic are neural networks with two hidden layers with size 64 and Tanh as the activation function, and the discriminator is a neural network with two hidden layers with size 100 and ReLU as the activation function. We use ADAM to update the imitation learning model and Stochastic Gradient Descent method (SGD) to update the confidence. We implement our method in the PyTorch framework [1]. We train each algorithm 10 times with different random seeds, and record how the expected return and the standard deviation varies during training. While testing the return, we run the algorithm for 100 episodes. While implementing Eqn. (11), we normalize $\beta$ so that their mean value is 1, i.e. for the first part of Eqn. (11), we use $\sum_{(s,a)\in\Xi} \left( -\frac{n\beta(s,a)}{\sum_{(s,a)\in\Xi}\beta(s,a)} \right) \log(D(s,a))$, where $n$ is the number of state-action pairs in $\Xi$. All the experiments in all environments are run on one Intel(R) Xeon(R) Gold 6244 CPU @ 3.60GHz with 10G memory.

### B.1.2 Environment

**Reacher.** In the Reacher environment, the agent is an arm with two links and one joint, and the end effector of the arm is supposed to reach a final location. Each step, the agent is penalized for the energy cost and the distance to the target.

We collect 200 trajectories in total for training, where each trajectory has 50 interaction steps. 5% of the trajectories are annotated with rankings. We collect 5 trajectories for testing. We run the experiment for 5 runs and compute the mean and the standard deviation of the expected return. The average time for each run is 1,291s.

**Ant.** In the Ant environment, the agent is an ant with four legs and each leg has two links and two joints. Its goal is to move forward in the x-axis direction as fast as possible. Each step, the agent is rewarded for moving fast in the x-axis direction without falling down, while it is penalized for the energy cost. If the ant fails to stand, the trajectory will be terminated.

We collect 200 trajectories in total for training, where each trajectory has at most 1000 interaction steps. 5% of the trajectories are annotated with rankings. We collect 5 trajectories for testing. We run the experiment for 5 runs and compute the mean and the standard deviation of the expected return. The average time for each run is 17,140s.

**Simulated Robot Arm.** In this environment, there is a Franka Panda Arm that is supposed to pick up a bottle, avoid the obstacle, and put the bottle on a target platform. Each step, the agent is penalized for the energy cost and the distance to the target. If the agent drops the bottle or hits the target, it will receive a large negative reward and the trajectory will be terminated. If the agent succeeds to make the bottle stand on the target, the trajectory will be terminated too, so that the arm will no longer receive penalization. The reachable region of the arm is $[0.20, 0.80]$ in x-axis, and $[-0.35, 0.35]$ in y-axis. The initial position of the bottle is sampled in $[0.68, 0.72] \times [-0.05, 0]$ and the initial position of the target is sampled in $[0.28, 0.32] \times [-0.32, -0.28]$. The action space is the velocity of the end-effector, and the maximum velocity is 1 in each direction.

We collect 200 trajectories in total for training, where each trajectory has at most 2000 interaction steps. 5% of the trajectories are annotated with rankings. We collect 5 trajectories for testing. We run the experiment for 5 runs and compute the mean and the standard deviation of the expected return. The average time for each run is 44,731s.

**Real Robot Arm.** In this environment, we use a real UR5e robot arm in a similar settings as the simulation environment.

We collect 200 trajectories in total for training, where each trajectory has at most 2000 interaction steps. 5% of the trajectories are annotated with rankings. We collect 5 trajectories for testing. We run

the experiment for 5 runs and compute the mean and the standard deviation of the expected return. The average time for each run is 141,699s.

### B.1.3 Evaluation Metrics

To evaluate the proposed CAIL and other methods, we use the expected return for all the environments, which is the discounted cumulative reward of a trajectory. For the Reacher and the Ant environments, we use the reward function in their original implementation in Gym[2]. For the Simulated Robot Arm and the Real Robot Arm environments, we define a reward as follows: Assume that the action of the robot arm (the velocity of the end-effector) is $a$, the distance between the bottle and the target is $d$, the distance between the bottle's initial position and the target is $d_{\text{init}}$, then at each step, the robot will receive a reward of $-\frac{0.02a}{d_{\text{init}}^2} - 0.05d$. If the robot drops the bottle or the obstacle is moved, the robot will receive a reward of $-2000$ and the trajectory will be terminated. In the robot arm environments (both simulated and real), we also use the success rate as another metric to evaluate the rate that the robot arm successfully moves the bottle to the goal area without colliding with the obstacle.

### B.2 Results

Table 1: The converged expected return of all the methods in Mujoco Reacher and Ant, Simulated Franka Panda Robot Arm, and the Real UR5e Robot Arm environments. We provide numerical results for a clearer comparison.

| Method | Reacher | Ant | Simulated Robot | Real Robot |
|--------|---------|-----|-----------------|------------|
| CAIL | **-7.816**±1.518 | **3825.644**±940.279 | **-62.946**±50.644 | **-34.330**±1.242 |
| 2IWIL | -23.055±3.803 | 3473.852±271.696 | -120.622±122.787 | -52.445±7.182 |
| IC-GAIL | -55.355±5.046 | 1525.671±747.884 | -349.511±342.597 | -550.235±657.838 |
| AIRL | -25.922±2.337 | 3016.134±1028.894 | -236.953±230.495 | -597.819±752.149 |
| GAIL | -60.858±3.299 | 998.231±387.825 | -527.604±452.379 | -532.854±664.415 |
| T-REX | -66.371±21.295 | -1867.930±318.339 | -1933.944±380.834 | -2003.672±32.771 |
| D-REX | -78.102±14.918 | -2467.779±135.175 | -1817.239±481.672 | -1538.100±703.266 |
| SSRR | -70.044±14.735 | -105.346±210.837 | -2077.616±58.764 | -2154.214±168.086 |
| Oracle | -4.312 | 4787.233 | -35.362 | -31.056 |

**Numerical Comparison.** We provide the numerical comparison of CAIL and the baseline methods in Table 1. The results correspond to the results in Fig. 2 in the main text. We can observe that CAIL outperforms all the baseline methods in all the environments and the margin between CAIL and the best-performing policy is much closer than the margin between baseline methods and the best-performing policy.

Table 2: Success rate (%) among 100 trials of all the methods in the simulated and real robot environments.

| Method | CAIL | 2IWIL | IC-GAIL | AIRL | GAIL | T-REX | D-REX | SSRR |
|--------|------|-------|---------|------|------|-------|-------|------|
| Simulated Robot | 100 | 100 | 81 | 87 | 31 | 0 | 0 | 0 |
| Real Robot | 100 | 83 | 7 | 33 | 20 | 0 | 0 | 0 |

**Success Rate.** We report the success rate among 100 trials of different methods in Table 2. We observe that for both simulated and real robot environments, CAIL achieves the highest success rate. Though 2IWIL also achieves a high success rate; however, it induces trajectories with longer detour and thus has lower expected return.

**Ablating the size of ranking dataset** We provide results of CAIL and the compared methods including 2IWIL, IC-GAIL, and T-REX with varying levels of supervision. We do not include GAIL, AIRL, D-REX, and SSRR in this ablation since they do not require any supervision. We conduct experiments in the Reacher environment, and vary the ratio of demonstrations labeled with ranking. The agents are provided with 200 trajectories, and the ratios of labeled demonstrations are 1%, 2%,

---

[2]https://github.com/openai/gym

Table 3: The performance with respect to the size of the ranking dataset.

| Label Ratio | 1% | 2% | 5% | 10% | 20% | 50% | 100% |
|---|---|---|---|---|---|---|---|
| 2IWIL | $-33.5 \pm 4.9$ | $-34.4 \pm 3.2$ | $-23.3 \pm 4.1$ | $-27.7 \pm 6.7$ | $-24.5 \pm 3.0$ | $-30.0 \pm 2.7$ | $-25.2 \pm 6.9$ |
| IC-GAIL | $-56.4 \pm 10.1$ | $-53.7 \pm 4.0$ | $-61.0 \pm 5.0$ | $-54.1 \pm 6.0$ | $-58.8 \pm 3.4$ | $-44.6 \pm 8.3$ | $-57.1 \pm 3.7$ |
| T-REX | $-83.7 \pm 18.6$ | $-85.8 \pm 15.3$ | $-82.3 \pm 10.2$ | $-73.2 \pm 21.6$ | $-91.8 \pm 15.4$ | $-38.6 \pm 35.8$ | $-27.2 \pm 37.2$ |
| CAIL | $\mathbf{-8.0 \pm 2.4}$ | $\mathbf{-8.7 \pm 3.6}$ | $\mathbf{-7.3 \pm 2.0}$ | $\mathbf{-8.1 \pm 2.9}$ | $\mathbf{-7.1 \pm 1.7}$ | $\mathbf{-7.5 \pm 2.3}$ | $\mathbf{-7.8 \pm 3.0}$ |

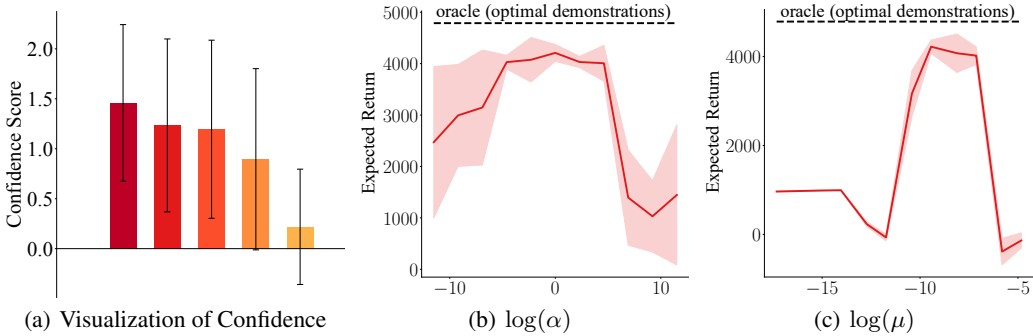

(a) Visualization of Confidence    (b) $\log(\alpha)$    (c) $\log(\mu)$

Figure 1: (a) The visualization of confidence for demonstrations drawn from policies with different optimality. There are 5 policies with different optimality, where the darker color means the policy has higher expected return. (b-c) The expected return with varying hyper-parameters $\alpha$ and $\mu$.

5%, 10%, 20%, 50%, 100%. The average trajectory rewards and the standard deviations are shown in the Table 3. CAIL outperforms all the other methods with a large gap in all settings, even in the setting with only 1% labeled demonstrations, i.e., only two trajectories are labeled, which is the minimum label we can have for ranking. 2IWIL and IC-GAIL, however, do not perform well, and there is no clear increase of performance as the label ratio increases. This is because what they need is labeled confidence, which is a much stronger type of supervision than ranking. The confidence cannot be accurately recovered when only given rankings. T-REX does not perform well either, but it is getting better as the ratio of labels increases. This experimentally proves that T-REX needs much more data than CAIL to learn a reward function and CAIL can use the demonstrations more efficiently.

**Visualization of the Confidence** In our experiments, we have 5 sets of demonstrations collected from 5 different policies, where each set of demonstrations has different average returns. So we can learn different average confidence values for each different set of demonstrations. The larger the average return, the larger the average confidence. In our framework, we learn a confidence for each state-action pair. We visualize the un-normalized confidence learned by CAIL of these 5 set of demonstrations in Fig. 1(a), where the darker color means the demonstrations have higher expected returns. We observe that the darker color bar has higher confidence, which indicates that CAIL-learned confidence matches the optimality of the demonstrations.

**Hyper-parameter Sensitivity.** We investigate the sensitivity of hyper-parameters including the two learning rates $\alpha$ and $\mu$. We aim to demonstrate two points: (1) The proposed approach can work stably with the hyper-parameters falling into a specific range; (2) If the hyper-parameters are too large or too small, the performance can drop, which means that tuning the two hyper-parameters are necessary for the performance of our algorithm. We conduct experiments in the Ant environment. As shown in Figure 1(b) and 1(c), the proposed approach work stably with $\alpha$ in the range $[10^{-3}, 100.0]$, and with $\mu$ in the range $[3 \times 10^{-5}, 3 \times 10^{-4}]$. When $\alpha$ and $\mu$ are too large or too small, the performance drops. The observations demonstrate the two points introduced above.

**Videos for Real Robot.** We show the videos of experiments in the real UR5e robot arm environment in the file 'robot_video.mp4' in the supplementary materials.

# References

[1] Adam Paszke, Sam Gross, Francisco Massa, Adam Lerer, James Bradbury, Gregory Chanan, Trevor Killeen, Zeming Lin, Natalia Gimelshein, Luca Antiga, et al. Pytorch: An imperative style, high-performance deep learning library. *arXiv preprint arXiv:1912.01703*, 2019.