# OpenReview forum: "Confidence-Aware Imitation Learning from Demonstrations with Varying Optimality"
_NeurIPS.cc/2021/Conference — NeurIPS 2021 Poster_

### Official Review · Reviewer_9kz8 · 2021-07-01

**Rating:** 6
**Confidence:** 4

**Summary:**

The paper proposes Confidence-Aware Imitation Learning (CAIL) that learns from sub-optimal demonstrations collected by different policies. The main idea is to train an imitating policy from demonstrations weighted by an estimated confidence scores, while the confidence scores are simultaneously estimated by using the imitating policy and ranked demonstrations. The paper also presents the convergence proofs of the algorithm. Experiments on simulated and real-world environments indicate that CAIL is more effective at learning from sub-optimal demonstrations when compared to existing algorithms.

**Limitations And Societal Impact:**

The main limitation of the proposed framework is that it requires an additional demonstration dataset for evaluation. This requirement is mentioned in the paper, but not explicitly as a limitation in Section 6. I suggest having a sentence in Section 6 to note this limitation.

**Main Review:**

In short, the paper presents an improvement over prior works and shows that the new algorithm works empirically well on simulated and real-world problems. However, the weaknesses of the paper are the technical quality and clarity. I currently rate the paper as a weak-rejection unless these weaknesses are addressed.

- Originality:

The paper presents a new idea to learn from sub-optimal demonstrations. This idea is a combination of two ideas: imitation learning with confidence scores [1] and imitation learning with ranked demonstrations [2]. The paper clearly mention these prior works and differentiate itself that the idea is different mainly because [1,2] did not jointly learn the confidence/ranking and policy. I view this as a fair improvement over the prior works.

- Quality:

The idea of the framework is sounded. I like that the framework is general and allows changing the loss functions for inner learning and outer learning. The analyses in Section 4.2 are also valuable to show that the framework can learn the optimal policy under some conditions. (I did not carefully check the proofs in the appendix, but the proofs seem to follow a standard convergence proof for smooth objective functions.) However, I could not understand the practical implementation in Section 4.3 and have several questions and comments.

1. In [1], the confidence is constrained to be a probability simplex that is normalized. However, the confidence in CAIL is unnormalized given Figure 1(a) in the appendix, and the upper-bound of the confidence is not defined. This unbounded $\beta$ should affect the gradient of AIRL loss, i.e., the first term can have a high magnitude from $\beta$ and dominates to second term in equation 11. How does CAIL handle this issue?

2. The inner loss $\mathcal{L}_{in}(s,a; \theta, \beta)$ is defined as a function of $(s,a)$, but the AIRL loss does not depend on a specific $(s,a)$ because the loss computes the expectation over state-action distributions. How does this affect the theoretical analyses?

3. Without defining $\beta$ to be a probability simplex as done in the prior work, the optimal solution in equation 1 is $\beta(s,a) = \infty$ which is not meaningful. While this is a minor point since the algorithm does not use equation 1, an important quantity such as the theoretically-optimal confidence should be more meaningful.

4. Figure 1(a) in the appendix plots the confidence of different policies, but $\beta$ is a function of state-action pair. Could the authors clarify how they compute the confidence in Figure 1(a)? Does the plot show the average of $\beta(s,a)$ in each trajectory?

- Clarity:

The writing is mostly clear. Though, there are some parts that could be improved:
1. The word "varying optimality" can be misunderstood as learning from multiple expert policies as considered by [3] where demonstrations in the dataset are optimal w.r.t. different rewards. However, CAIL assumes there is only one reward function and only one optimal policy. I think "varying optimality" is too ambiguous.

2. Theorem 2 overclaims that the solution is the optimal confidence. The theorem guarantees only the convergence to a local solution where the gradient is zero. Since it is not assumed that $\mathcal{L}_{out}$ is convex w.r.t. $\beta$, the solution is not always the optimal confidence. The claim from Theorem 2 should be revised.

3. There is an inconsistent notation: Equation 2 uses $\beta(s,a) p_{\pi^d}(s,a)$ to denote the weighted sampling distribution, but equation 11 uses $\beta(s,a) \pi^d$.

4. There is a slight gap between line 146 where the optimal confidence is defined and line 158 where the objective to learn the confidence is defined. That is, are equation 1 and equation 3 equivalent? If they are, it should be shown. If they are not, it should be explained why equation 3 is a good surrogate to equation 1 and how the choice of surrogate (i.e., the outer loss) affect the solution.

- Significance:

The paper is significant in terms of algorithmic improvement and practical demonstrations.



[1] Yueh-Hua Wu, Nontawat Charoenphakdee, Han Bao, Voot Tangkaratt, and Masashi Sugiyama. Imitation learning from imperfect demonstration. ICML, 2019

[2] Daniel Brown, Wonjoon Goo, Prabhat Nagarajan, and Scott Niekum. Extrapolating beyond sub optimal demonstrations via inverse reinforcement learning from observations. ICML, 2019.

[3] Karol Hausman, Yevgen Chebotar, Stefan Schaal, Gaurav Sukhatme, and Joseph J. Lim. Multi-Modal Imitation Learning from Unstructured Demonstrations using Generative Adversarial Nets. NeurIPS, 2017.


**Time Spent Reviewing:**

Around 4 hours.

---

> ### Author Response · Authors · 2021-08-10
> **Response**
>
> **TECHNICAL CONCERNS:**
>
>
> **1. UNNORMALIZED CONFIDENCE IN FIG. 1(A) in APPENDIX:**
> In Fig. 1(a) in the Appendix, we wanted to show that the learned confidence is consistent with the optimality of the demonstrations, so we just plotted the unnormalized confidence values. However, before multiplying the loss with the confidence weight, we do normalize it to set the mean equal to one. We will clarify this point in the revision.
>
>
> **2. INNER LOSS DEFINED AS A FUNCTION OF (s,a):**
> The inner loss in Eqn. (4-7) and the inner loss in the theoretical analyses are all computed as the expectation over state and actions, and are not a function of a single state-action pair $(s,a)$. When conducting experiments, we use mini-batches of data for the update. We will clarify this point in the revision by expanding the definition of the inner and outer loss and emphasizing its dependence on the expectation over state and actions.
>
>
> **3. DEFINING $\beta$ AS A PROBABILITY SIMPLEX:**
> We implicitly constain $\beta$ in the inline equation $p_{\pi_\text{new}}(s,a)=\beta(s,a)p_{\pi^d}(s,a)$ on Line 153 since both $p_{\pi_\text{new}}(s,a)$ on the left and $p_{\pi^d}(s,a)$ on the right should satisfy the requirement of the probability distribution. We agree that we should more clearly show this and will clarify this point in the revision.
>
>
> **4. CLARIFICATION ON COMPUTING CONFIDENCE IN FIG. 1(a):**
> In our experiments, we have 5 sets of demonstrations collected from 5 different policies, where each set of demonstrations has different average returns. So we can learn different average confidence values for each different set of demonstrations. The larger the average return, the larger the average confidence. In our framework, we learn a confidence for each state-action pair. Fig. 1(a) in the Appendix shows the average and the standard deviation of the learned confidence for all the state-action pairs in each set of demonstrations. We use the darker color to indicate that the corresponding set of demonstrations has a higher average return. Our goal for this figure is to show that the set of demonstrations with a higher return also has higher average learned confidence.
>
>
>
> **CLARITY CONCERNS:**
>
> **1. THE TERM ‘VARYING OPTIMALITY’:**
> We agree that ‘varying optimality’ can be confusing especially compared to how it was used in prior work. We hope to clarify this in the final revision. Our convention was that optimality would measure how good a demonstration is with respect to the reward function. We used the term ‘optimality’ as a continuous value where demonstrations with higher reward also have higher optimality, but as the reviewer explains there exists a single optimal reward structure.
>
>
> **2. THEOREM 2: CONVEXITY ASSUMPTION:**
> We agree with the reviewer that the algorithm is guaranteed to converge to the optimal confidence only when $\mathcal{L}_\text{out}$ is convex with respect to $\beta$. We will revise the theorem to emphasize the convexity assumption for convergence and make it more rigorous.
>
>
> **3. INCONSISTENT NOTATION IN EQN. (2):**
> Thanks for bringing this to our attention. We will correct this in the revision.
>
>
> **4. GAP BETWEEN THE DEFINITION OF OPTIMAL CONFIDENCE ON LINE 146 and 158:**
> Eqn. (1) is the definition of the optimal confidence $\beta^*$, which is defined as the confidence to derive the best expected return. Eqn. (3) estimates this optimal confidence $\beta^*$ based on the evaluation loss $\mathcal{L}_\text{out}$ and the evaluation data to compute the outer loss value. We plan to use $\beta^*_\text{out}$ to instead refer to the confidence value in Eqn. (3) emphasizing the reliance on the evaluation loss and data. If $\mathcal{L}_\text{out}$ is defined to output lower loss for a policy with higher expected return and the evaluation data is sufficient, the $\beta^*$ in Eqn. (1) and (3) are nearly the same. We will re-emphasize the requirement of the evaluation dataset (rankings in our experiments) in Section 6.
>
> **ADD THE LIMITATION TO SECTION 6:**
> We will revise section 6 according to your suggestions.

---

> > ### Comment · Reviewer_9kz8 · 2021-08-16
> > **Reply**
> >
> > Thank you for the response. The response clarifies and addresses the weaknesses I commented in the review. I think this will be a good paper once the clarifications are added to the paper. I have increased the rating in my review.

---

### Official Review · Reviewer_1GZR · 2021-07-10

**Rating:** 7
**Confidence:** 3

**Summary:**

This paper investigates imitation learning under the setting where there are non-interactive state-action demonstrations with varying optimality, and among them a small percentage has trajectory-level ranking annotations. The authors propose an imitation learning framework that can leverage all demonstrations, although some of them are suboptimal or even malicious, and potentially outperform the demonstrators. The idea is to formulate learning a state-action confidence function (which weights demonstrations) as a bi-level optimization problem. This closely ties IRL and ranking matching. The authors prove the convergence rate of the resulting algorithm -- CAIL -- and show empirically both in simulation and in real-word that CAIL outperforms recent algorithms designed for similar settings.


**Limitations And Societal Impact:**

yes

**Main Review:**

Strengths:

Using bi-level optimization to learn the confidence function in IRL seems new. Compared with the two-step approaches, e.g. I2WIL and T-REX, that first learns a confidence function then applies IRL, the bi-level optimization formulation tightly connects the IRL with ranking matching to effectively leverage all demonstrations. Even though bi-level optimization is harder to solve than the two-step formulations, empirically it demonstrates better or on-par performance.
Encourage results:

In the experiments, the authors compare CAIL with a range of state-of-the-art algorithms and demonstrate encouraging results, especially in Fig 3.


Weakness:

One of the three claimed main contributions in the paper is proving the convergence rate of the resulting algorithm CAIL. However, the bi-level formulation , i.e., $\min_\beta L_{out}(\theta^*(\beta))$ where $\theta^*(\beta) = \arg\min_\theta L_{in}(\theta, \beta)$, and analysis seems to be standard.  If the authors can explain the challenges and novelties in the analysis, that would be great.

Regarding writing, the paper is overall easy to follow. However, I think the framework proposed in Section 4 is a bit abstract and readers may easily get lost. For example, in Section 4, L_out (the ranking loss) appears in Eq. (3), but it’s unclear how L_out is a function of the parameters of any imitation learning algorithm. Therefore, it might be helpful to discuss that L_out relies on an estimate of the reward function in Section 4.
Furthermore, I found the following points confusing:
- Eq. (1) and (3) both define beta*. But they are different. For instance, Eq. (3) is sample-based ranking loss.
- If Eq. (4) and Eq.(6) are applied until convergence, what is the difference between them?

Regarding experiments, in the simulated Franka Panda Arm and real robots experiments, CAIL does not seem to out-perform 2IWIL, which is arguably a simpler approach as during policy optimization the confidence function is fixed. In the Reacher task (Fig 2e) and Ant task (Fig 3c), it’s interesting that the performance of 2IWIL and ARIL decrease with increasing interaction steps; whereas CAIL improves monotonically. I wonder whether authors have intuitions for this phenomenon. If the highest expected return is reported instead of the last policy expected return, how would Fig 3a and Fig 3b look like? In addition, interestingly, in Fig 3c, with pure suboptimal demonstrations, AIRL can still learn a policy that has expected return close to the best policy learned by CAIL with 5e6 iterations steps. It will be helpful if the authors can provide the expected return of the experts.

It’s also worth noting that the optimal confidence function \beta* is defined by an empirical risk minimizer (Eq. 13). It’s unclear to me why the bi-level optimization approach is able to learn with a smaller amount of data than 2IWIL and T-REX.

Other comments:
- p-values are used to show statistical significance in the experiments. What are the null hypotheses and tests?
- In P4 L 153, \beta(s,a) is used to define the state-action probability of \pi_new. Is certain normalization property enforced for function \beta?
- In P3 L91, it’s mentioned that the paper focuses on the offline learning setting. However, the instantiation of CAIL (Section 4.3) needs interactions with the environment.


**Time Spent Reviewing:**

4

---

> ### Author Response · Authors · 2021-08-10
> **Response**
>
> **DIFFERENCE BETWEEN EQN. (4) and (6) AND NOVELTY IN CONVERGENCE PROOF:**
> Eqn. (4) is a pseudo-update step, which does not really change the value of $\theta$ but just computes a proxy of updated $\theta$ for computing the gradients of $\beta$ in Eqn. (5). Eqn. (6) updates $\theta$, which changes the value. We can see that the gradient of $\theta$ in Eqn. (4) is based on the $\beta_\tau$, while the gradient of $\theta$ in Eqn. (6) is based on the updated $\beta_{\tau+1}$. $\beta_{\tau+1}$ is a more recent and accurate estimate of the confidence than $\beta_\tau$, so the gradient of $\theta$ in Eqn. (6) shows a better update direction of $\theta$ and can potentially accelerate the convergence. So we use Eqn. (6) to actually update $\theta$.
>
> Using an additional update (i.e., Eqn. (6)) to update $\theta$ is also the main difference between our bi-level optimization and conventional bi-level optimization.  Conventional bi-level optimization alternatively computes gradients and updates $\theta$ and $\beta$, which will use our pseudo-update in Eqn. (4) to ultimately change the value of $\theta$ and do not have Eqn. (6).
>
> We agree that the core idea of the proof still remains the same as the proofs for conventional bi-level optimization methods. The key difference/novelty of this proof lies in all the inequalities including the pseudo-update parameter $\theta’_{t+1}$.  We will edit the contributions of the paper to emphasize the advances of this proof compared to existing proofs of conventional bi-level optimization.
>
>
> **L_out AS A FUNCTION OF PARAMETERS OF IMITATION LEARNING ALGORITHMS:**
> $\mathcal{L}_\text{out}$ evaluates the performance of the underlying imitation learning algorithm with respect to the reward, so $\mathcal{L}_\text{out}$ maps the imitation learning model parameters $\theta$ to a value related to the expected return of the imitation learning policy. We agree the reliance of $\mathcal{L}_\text{out}$ on the estimate of the reward is a bit abstract, and we hope to clarify this in the revision by providing a concrete example in the text.
>
>
> **DIFFERENCE BETWEEN $\beta^{*}$ in EQN (1) and (3):**
> Eqn. (1) is the definition of the optimal confidence $\beta^*$, which is defined as the confidence to derive the best expected return. Eqn. (3) estimates the optimal confidence $\beta^*$ based on the evaluation loss $\mathcal{L}_\text{out}$ and the evaluation data to compute the outer loss value. We apologize for the confusion, and we will clarify the difference by specifying $\beta^*$ in EQN. (3) as $\beta^*_\text{out}$ representing its dependence on evaluation loss and data. If $\mathcal{L}_\text{out}$ is defined to output a lower loss for a policy with higher expected return and the evaluation data is sufficient, the $\beta^*$ in Eqn. (1) and (3) will nearly be the same.
>
>
> **PERFORMANCE OF CAIL AND 2IWIL IN ROBOT ENVIRONMENTS:**
> In the simulated robot environment, the performance gap between 2IWIL and CAIL is small and not statistically significant. The reason is that the simulated robot environment is a cleaner environment compared to  the real robot environment. This makes the task easier and reduces the performance gap.
>
> In the real robot environment, Fig. 2(h) shows a small margin between CAIL and 2IWIL. However, we would like to emphasize that, the variance of 2IWIL is small in the real robot environment and CAIL actually outperforms 2IWIL statistically significantly. The appearance of a small gap in Fig. 2(h) could be due to the way we are visualizing the results as the gap between CAIL and some of the other methods are much larger; thus making it difficult to visualize the statistical significance between CAIL and 2WIL. Also the videos in our supplementary show that 2IWIL learns a policy with more detours than CAIL.
> We have now also computed the oracle performance, and the highest return in the demonstrations is about -34 for both the simulated and the real robot experiments. As shown in our numerical results in Table 1 of the Appendix, in the simulated robot environment, the margin between the best demonstration and CAIL is much smaller than the margin between CAIL and 2IWIL. In the real robot environment, CAIL achieves nearly the same reward as the best demonstration while 2IWIL has a non-negligible margin compared to the best demonstration. These results demonstrate that the performance gain of CAIL is significant.
>
>
> **INTUITION BEHIND DECREASING PERFORMANCE OF 2IWIL AND AIRL:**
> For AIRL, we compare Fig. 2(h) and Fig. 3(c) to explain this performance decrease, where both experiments are conducted in the Ant environment. We observe that in both figures, the performance of AIRL first increases to about 2700. This is because even though the demonstrations are suboptimal, there are potentially optimal state-action transitions in the suboptimal demonstrations leading to the initial high performance. However, at this early training stage, the AIRL model does not converge yet and the model parameters can still change rapidly. After training a sufficient number of steps, the AIRL model will converge to the average return of the demonstrations which includes the useful transitions in the suboptimal demonstrations at the early stages as well as some of the less useful transitions that might be observed later. We observe the converging phenomenon in both Fig. 2(h) and Fig. 3(c). The reason for the stable performance in Fig. 2(h) and the performance decrease in Fig. 3(c) is that the average return of demonstrations for Fig. 3(c) is much lower than Fig. 2(h).
> We also argue that in Fig. 3(c) the performance of 2IWIL is not clearly decreasing but is instead unstable.
>
> The average return of the demonstrations is 2397.93 and the highest return in the demonstrations is 3739.91.
>
> Comparing the convergence performance, CAIL converges to a policy with the expected return of about 3100, which is close to the highest return in the demonstrations and higher than the average return, while 2IWIL and AIRL converge to policies with the expected return of about 2000, which are even lower than the average return of the demonstrations. The margin between CAIL and oracle, which is about 650, is much smaller than the margin between AIRL/2IWIL and oracle, which is about 1750. This shows the significance of the improvement.
> Comparing the highest performance point, the highest point in the AIRL curve is about 2700 and the highest point of CAIL is about 3200. This means the margin between CAIL and the best-performing policy is much closer than the margin between AIRL and the best-performing policy.
>
> In Fig. 2(a) and 2(b), the experiments are conducted in the Reacher environment. We double-check the results and find there is no such performance decrease phenomenon in the Reacher environment.
>
>
> **SMALLER AMOUNT OF DATA FOR CAIL COMPARED TO 2IWIL AND T-REX:**
> Compared to T-REX, T-REX needs to learn a reward function over all state-action pairs to learn a well-performing policy in its following reinforcement learning steps, while CAIL only needs to learn the confidence value for the state-action pairs in the demonstration set, which is a much smaller space than the whole state-action space. Thus, CAIL needs much less data to cover a much smaller space.
>
> Compared to 2IWIL, the strength of CAIL is not on the amount of data but using rankings of demonstrations instead of the confidence metric. Confidence is a stronger annotation than rankings since rankings can be derived from confidence but the reverse is not true.
>
>
> **NULL HYPOTHESIS AND STATISTICAL TESTS:**
> The null hypothesis is the performance of CAIL is equal to or smaller than the baseline methods. We use the student’s t-test.
>
>
> **NORMALIZATION PROPERTY ENFORCED ON $\beta$:**
> We implicitly mention the normalization property of $\beta$ in $p_{\pi_\text{new}}(s,a)=\beta(s,a)p_{\pi^d}(s,a)$ on Line 153 since both $p_{\pi_\text{new}}(s,a)$ on the left and $p_{\pi^d}(s,a)$ on the right should satisfy the requirement of the probability distribution. In our experiments, we sample (s,a) pairs in the demonstrations to estimate the distribution, so while implementing Eqn. (11), we normalize $\beta$ so that their mean value is 1, i.e. for the first part of Eqn. (11), we use $\sum_{(s,a)\in\Xi}\left(-\frac{n\beta(s,a)}{\sum_{(s,a)\in\Xi}\beta(s,a)}\right)\log(D(s,a))$, where $n$ is the number of state-action pairs in $\Xi$.
> We will clarify this normalization in the revision.
>
>
> **OFFLINE IMITATION LEARNING SETTING:**
> The offline learning here means that we use a set of offline demonstrations and do not collect more demonstrations in an online or interactive fashion. We do not mean that the imitation learning algorithm is offline. We will clarify this point in the revision.

---

> > ### Comment · Reviewer_1GZR · 2021-08-16
> > **Two more questions**
> >
> > Thank you very much for the helpful response. I have two more questions.
> >
> > (1) In the main text, in L187, you mentioned “after convergence”. I was confused that if the gradient descent is run until convergence in iteration t, in iteration t+1, the gradient used in Eq (4) would become zero and thus nothing to update. Is this correct?
> >
> > (2) I think you reference to the figures in the paragraph “INTUITION BEHIND DECREASING PERFORMANCE OF 2IWIL AND AIRL” in your reply was incorrect since Fig 2(h) is not for Ant and Fig. 2(a) and 2(b) are not for Reacher. Would you please clarify that? Thanks!

---

> > > ### Author Response · Authors · 2021-08-18
> > > **Response to additional questions**
> > >
> > > Thank you for reading our response and we address the further questions as follows.
> > >
> > > (1) The ‘After convergence’ on L 187 means that we update $\theta_t$ until convergence according to Eq. (6) at time step $\tau$ and then we get $\theta_{\tau+1}$. In this process $\beta$ is fixed as $\beta_{\tau+1}$. $\theta$ converges when
> > > $\triangledown_\theta\mathcal{L}_\text{in}(s,a;\theta_t,$
> > >
> > > $\beta_{\tau+1})=0$
> > > . The ‘After convergence’ here does not mean the whole optimization process converges. Eq. (4) is about the pseudo-update and is used for updating $\beta$, which is not related to the ‘convergence' on L 187.
> > >
> > > Also, we want to clarify that the optimization process in Eq. (4)(5)(6) is too time-consuming and we use the one-step update version in Eq. (7) in our method and in our experiments.
> > >
> > > (2) We apologize for the typos in the fig name. In the answer **INTUITION BEHIND DECREASING PERFORMANCE OF 2IWIL AND AIRL**, the 'fig 2(h)' should be corrected as 'fig 2(f)' to point at the Ant results. The 'fig 2(a) and 2(b)' should be corrected as 'fig 3(a) and 3(b)', which are done in the Reacher environment.

---

### Official Review · Reviewer_g7rL · 2021-07-15

**Rating:** 8
**Confidence:** 3

**Summary:**

* This paper proposes a novel imitation learning algorithm, Confidence-Aware Imitation Learning (CAIL), to learn policy from suboptimal data.
  * In particular, this proposed algorithm iteratively does two gradient updates:
    1. learns a function, \beta, that maps (state, action) to a confidence score using an evaluation dataset (e.g., a limited amount of partial rankings) with an outer loss.
    2. learn the imitation learning model parameters \theta by reweighting the full IL dataset (could be suboptimal, without labeled rankings) using \beta, and optimizing for an inner loss.
  * The authors proposed a convergence rate of the proposed method as a bi-level optimization problem (Theorem 2).

* The authors conduct a comprehensive evaluation between the proposed algorithm with a set of baseline algorithms.
  * Domains: MuJoCo reacher, MuJoCo ant, Franka Panda simulation, UR5e real robot.
  * Baselines:
    * Standard imitation learning algorithms: GAIL, AIRL.
    * Confidence-based methods: 2IWIL, ICGAIL.
    * Ranking-based methods: T-REX, D-REX, SSRR.
  * Results:
    * Reacher: CAIL outperformed other methods in expected return.
      * All comparisons (between CAIL and another method) are statistically significant.
    * Ant: CAIL outperformed other methods in expected return.
      * The p-value between CAIL and the closest baseline method, 2IWIL, is 0.1405.
    * Franka Panda simulation: CAIL outperformed other methods in expected return.
      * The p-value between CAIL and the closest baseline, 2IWIL, is 0.0974.
    * UR5e real robot: CAIL outperformed other methods in expected return.
      * All comparisons (between CAIL and another method) are statistically significant.
    * Demonstrations with Different Optimality
      * CAIL consistently outperforms or performs comparably to other methods with demonstrations at different optimality.
      * CAIL performs more stably while the baselines suffer from a performance drop at speciﬁc optimality levels.
    * Learning from Only Non-optimal Demonstrations
      * Ant: CAIL outperformed other methods in expected return.
        * All comparisons (between CAIL and another method) are statistically significant.


**Limitations And Societal Impact:**

* The main limitation has been discussed in Sec. Limitations and Future Work.

# Post rebuttal
Thank you for the clarification and additional explanations!
I have decided to keep my original score.

**Main Review:**


* The contribution is clear, and evaluation is comprehensive
  * The authors propose a novel IL algorithm that allows the robot to learn from suboptimal demonstrations. The authors show that the proposed method achieved a higher expected return in the learned policy than a set of baseline methods.

* One potential typo/mistake (in case it is truly a mistake, results might need to be rerun)
  * Eq. 14: In Eq. 14 2nd case, I think it should be `-\epsilon`?
    * I tried to plot the value of `RK[x,x,x,x]` as a function of `1[x>x](x-x)`, but it is not continuous. I think if you write `-\epsilon` in the 2nd case, it would be continuous?

* The writing is clear. Some comments:
  * Line 60: `with varying optimality`: when I first read this, I was confused. I thought `optimality` is a binary term, optimal or not. Later, I understood what the authors meant. I think maybe add some explanations or examples in Sec. 1 about this `varying`.
  * Line 178: `pseudo-update`: it might be very helpful to provide some insights into why this `pseudo-update` is necessary and why we cannot just simply use `\theta_t` in Eq. 5. My understanding is that it is required in the convergence proof?
  * Eq. 14: on the right of the 1st case, there is a missing `)`.
  * Eq. 14: on the left-hand side, the `;` and `,` are both used.
  * Line 312: `For Ant, the p-value between CAIL and the closest baseline method, 2IWIL, is 0.1405`: It would be helpful if any insights into this insignificance can be provided.
  * Line 335: `For the simulated robot arm environment, the p-value between CAIL and the closest baseline, 2IWIL, is 0.0974`: It would be helpful if any insights of this insignificance can be provided.
  * Line 343: `Demonstrations with Different Optimality`: I am not sure in which domain the authors evaluated this.

* I didn't check the proofs in the appendix.


**Time Spent Reviewing:**

6

---

> ### Author Response · Authors · 2021-08-10
> **Response**
>
> **TYPO IN EQN. (14):**
> There are two typos in Eqn. (14). One is that the right parenthesis is missing in the first line. The other is that the ‘+’ in the second line should be corrected as ‘-’. All our experiments were run based on the correct version of Eqn. (14).
>
>
> **THE TERM ‘OPTIMALITY’:**
> We use the term ‘optimality’ in a non-binary fashion. Specifically, optimality measures how good the demonstration is with respect to the reward function. It is a continuous value where demonstrations with higher reward also have higher optimality. We will clarify this in the revision.
>
>
> **INSIGHTS ON OUR PSEUDO-UPDATE:**
> The pseudo-update in Eqn. (4) uses the gradients of $\theta$ that are based on the confidence $\beta_\tau$ from the last step. The update in Eqn. (6) uses the gradients of $\theta$ that are based on the updated confidence $\beta_{\tau+1}$, which is a more accurate estimation of the confidence. Thus, the gradients of $\theta$ in Eqn. (6) show a better update direction of $\theta$ and can potentially accelerate the convergence. So we use the update in Eqn. (6) to update $\theta$ while use the pseudo-update in Eqn. (6) for the computation of the gradients of $\beta$.
>
>
> **INSIGHTS ON THE STATISTICAL SIGNIFICANCE RESULTS:**
> The lack of statistical significance in the Ant and simulated robot environments is caused by the fact that 2IWIL has a very high variance in these environments. 2IWIL assigns incorrect confidence to demonstrations and this causes its unstable performance: when sampling a batch of good state-action pairs, the performance increases while the performance decreases with a batch of bad state-action pairs. On the other hand, there is still a non-negligible gap between the mean performance between our approach (CAIL) and 2IWIL.
>
>
> **DOMAIN FOR EVALUATING `Demonstrations with Different Optimality’:**
> The domain we have used for this evaluation is the Reacher environment.

---

### Official Review · Reviewer_Tsdv · 2021-07-15

**Rating:** 7
**Confidence:** 3

**Summary:**

This paper considers the problem of offline imitation learning from demonstrations when these demonstrations may have been generated from a mixture of optimal, suboptimal, and even adversarial (i.e. worse than random) agents. To address this problem, the authors introduce Confidence-Aware Imitation Learning or CAIL, an imitation learning methodology where the policy is optimized jointly with a weight function that dynamically reweighs the "expert" demonstrations so that policy more closely imitates the high quality demonstrations. To enable the training of this weight function, CAIL requires, similarly to prior work, some additional supervision in the form of demonstration ranks, namely the quality of a small collection of demonstrations are ranked relative one another. Two theorems show that CAIL converges at a given rate under intuitive assumptions. Finally a collection of experiments show that, for both simulated environments and real-world robotic settings, CAIL outperforms competitive baselines.


**Limitations And Societal Impact:**

The noted limitation that you "still require that the dataset consists of demonstrations that encode useful knowledge for policy learning" is true but is a bit broad. I would prefer your limitations focus on concrete limitations of your study, e.g. (1) what about high-dimensional complex tasks?  (2) how do the assumptions of your proofs fail in practice? (3) how realistic/expensive is it to collect trajectory rankings in real settings? (4) how often do we really have large collections of mixed optimality demonstrations?

**Main Review:**

## Post-rebuttal response

I thank the authors for their helpful response to my, and other reviewer, comments. I found the additional ablation of the size of the ranking dataset to be particularly enlightening. I was positive about this paper before the rebuttal period and have not found any reason to change my opinion, I would be happy to see this paper accepted.

---

## Overall

While I do have some small concerns and questions (see below) I'm generally positive about this paper:

* Learning from suboptimal demonstrations is clearly an important problem although the requirement for additional ranking supervision is a bit limiting.
* CAIL's formulation as a bi-level optimization problem is interesting and the theoretical results are promising.
* CAIL substantially outperforms other ranking based methods across a number of diverse tasks and is competitive with confidence-based methods.

Baring any surprises or poor responses to my below points, I argue for the acceptance of this paper.

## Clean sensor modalities

Given B.1.1, it seems that, even for the real robotic experiments, that you use  clean, low-dimensional, inputs and relatively small neural models. My confidence in the generalizability of your experiments would be improved if there were, even preliminary, results for more complex settings (e.g. see any existing task in popular embodied simulators AI2-THOR, Habitat, iGibson, etc). Have you considered any such experiments?

## Ablating the size of the ranking dataset

While your Fig. 3 provides some interesting ablations for varying optimality, I am most interested in how the ranking and confidence based methods perform as the amount of additional supervision from ranks/confidence scores is varied. Would you have any additional results for this?

## Line-by-line comments/questions

In the following I give line-by-line comments and questions.

Line 46
 - I would suggest being more explicit about what is being ranked here. I.e. clarify that you require a dataset of ranked demonstrations.


Line 48-50
 - I found this a bit vague on my first reading, while you are technically agnostic to the imitation learning algorithm I would suggest grounding your introduction by describing how you actually achieve this (you hint at this in the next paragraph but I would mention explicitly that you use the reward function learned by AIRL to rank demonstrations and then compare these rankings against your gold labels).


Line 117
 - I'm not sure I understand what makes this return "expected". What is the expectation taken over when the demonstration is fixed and $\mathcal{R}$ is a function?


Line 127-128
 - In most cases I would expect demonstrations to be labeled with their source, i.e. demonstration $d_j$ came from annotator $i$. This additional label $i$ could be used as an additional feature for $\beta$ and could act as a strong signal for learning which demonstrations can be safely ignored. Have you considered this setting?


Line 177
 - I would suggest formally naming the $\beta$-functions so that you do not have to use somewhat awkward phrases like "value of confidence".


Line 178
 - I think it would be worth defining pseudo-update or giving some intution as to why you have named it this (as otherwise one has to wait until L185 before things start to become clear).


Line 190-192
 - While taking multiple gradient steps is a problem, it seems to me that the bigger problem is that you're, unless I'm mistaken, taking gradients of gradients of graidents of... This is quadratic in time but exponential in the parameter dimension.


Line 202
 - I'd suggest placing the footnote after the comma (i.e. \tau,\footnote{...}) as otherwise this looks like an exponent.


Line 254
 - You should define the indicator function here rather than waiting to L256.


Equation (14)
 - There are mismatched parentheses in the first $|...| > \epsilon$ statement.
 - Why not bring the negative sign into the difference on the first line of the equation?
 - I suspect there is something wrong with the second line of the equation. Note that the exxpression inside the maximum is always positive (as it's squared) so the max with 0 is unnecessary. This second line also seems to have some strange behavior, it becomes larger when the rankings agree but smaller when they don't?


Lines 256-259
 - I would appreciate a proof that RK is Lipschitz.


Line 310
 - How are you computing these p-values?
 - "The p-value" -> "the p-value"


Line 327
- "...where similar to before the..." -> "...where, similarly as before, the..."


Equation (8) in the appendix
- In the second to the last line, $\mu G$ should be $\mu C$.

**Time Spent Reviewing:**

4

---

> ### Author Response · Authors · 2021-08-10
> **Response**
>
> **ABLATING THE SIZE OF RANKING DATASET:**
> Here we provide results of CAIL and the compared methods including 2IWIL, IC-GAIL, and T-REX with varying levels of supervision. We do not include GAIL, AIRL, D-REX, and SSRR in this ablation since they do not require any supervision. We conduct experiments in the Reacher environment, and vary the ratio of demonstrations labeled with ranking. The agents are provided with 200 trajectories, and the ratios of labeled demonstrations are $1\\%, 2\\%, 5\\%, 10\\%, 20\\%, 50\\%, 100\\%$. The average trajectory rewards and the standard deviations are shown in the Table below. CAIL outperforms all the other methods with a large gap in all settings, even in the setting with only $1\\%$ labeled demonstrations, i.e. only two trajectories are labeled, which is the minimum label we can have for ranking. 2IWIL and IC-GAIL, however, do not perform well, and there is no clear increase of performance as the label ratio increases. This is because what they need is labeled confidence, which is much stronger supervision than ranking. The confidence cannot be accurately recovered when only given rankings. T-REX does not perform well either, but it is getting better as the ratio of labels increases. This experimentally proves that T-REX needs much more data than CAIL to learn a reward function and CAIL can use the demonstrations more efficiently. We will include this table in the Appendix.
>
>
> |Label Ratio | 1% | 2% | 5% | 10% |  20% |  50% | 100% |
> |:---:|:---:|:---:|:---:|:---:|:---:|:---:|:---:|
> |CAIL|$-8.0 \pm 2.4$  |  $-8.7\pm 3.6$ | $-7.3\pm 2.0$ | $-8.1\pm 2.9$ | $-7.1\pm 1.7$ | $-7.5\pm 2.3$ | $-7.8\pm 3.0$ |
> |2IWIL| $-33.5\pm 4.9$ | $-34.4\pm 3.2$ | $-23.3\pm 4.1$ | $-27.7\pm 6.7$ | $-24.5\pm 3.0$ | $-30.0\pm 2.7$ | $-25.2\pm 6.9$ |
> |IC-GAIL|$-56.4\pm 10.1$|$-53.7\pm 4.0$|$-61.0\pm 5.0$|$-54.1\pm 6.0$|$-58.8\pm 3.4$|$-44.6\pm 8.3$|$-57.1\pm 3.7$|
> |T-REX|$-83.7\pm 18.6$|$-85.8\pm 15.3$|$-82.3\pm 10.2$|$-73.2\pm 21.6$|$-91.8\pm 15.4$|$-38.6\pm 35.8$|$-27.2\pm 37.2$|
>
>
> **COST OF COMPUTING GRADIENTS OF GRADIENTS:**
> To update the high-dimensional parameter $\theta$ of the imitation learning model, we use the first-order gradient as shown in Eqn. (6), which costs the same as standard imitation learning algorithms.
>
> To update $\beta$, we use the $\beta$’s gradient of $\theta$’s gradient. However, $\beta$ is only a one-dimension scalar for each state-action pair and within each iteration of training, we only sample a mini-batch of thousands of state-pairs for update. Thus, within each iteration, the total dimension of $\beta$ is small and computing the gradient of gradient is not costly.
>
>
> **RK IS LIPSCHITZ: PROOF:**
> We consider the case where $\eta_{\xi_i}>\eta_{\xi_j}$. The case for $\eta_{\xi_i}<=\eta_{\xi_j}$ can be demonstrated similarly. We prove that RK is Lipschitz by definition. Let $z = \xi'_i-\xi'_j$, the derivative of RK is
> $$
> \triangledown RK_z=
> \begin{cases}
> 0 & z>\epsilon, \\\\
> \frac{1}{2\epsilon}(z-\epsilon) & -\epsilon\le z\le\epsilon, \\\\
> -1 & z<-\epsilon
> \end{cases}
> $$
> Thus, the second-order derivative of $RK$ satisfies that $|\triangledown (\triangledown RK_{z})_{z}|\le \frac{1}{2\epsilon}$. If we take $L>\frac{1}{2\epsilon}$, then $\forall z_1,z_2, |\triangledown(z_1)-\triangledown(z_2)|<L |z_1-z_2|$. This proves that RK is Lipschitz smooth.
>
>
> **MORE COMPLEX TASKS:**
> Due to the time limit, it is difficult to design tasks and run our method and all baselines in the suggested complex environments. However, we plan to add one of these environments in the revision.
>
>
> **THE TERM ‘EXPECTED RETURN’ ON LINE 117:**
> The term ‘expected’ here means computing the expectation of the reward of all time steps in one trajectory, but not computing the expectation across a set of trajectories. We will change this term for clarification in the revision.
>
>
> **TYPO IN EQN. (14):**
> There are two typos in Eqn. (14). One is that the right parenthesis is missing in the first line. The other is that the ‘+’ in the second line should be corrected to ‘-’.
>
>
> **ADDITIONAL LABELS FROM ANNOTATORS:**
> We do not consider the annotator label as an additional feature to predict $\beta$ in this paper and will consider this in our future work.
>
>
> **WRITING SUGGESTIONS:**
> Thank you for your suggestions including suggestions on narrowing down in our limitations and future work section. We will revise the paper following your feedback.

---

### Decision · Program_Chairs · 2021-09-27

**Decision:**

Accept (Poster)

**Comment:**

This paper considers imitation learning from demonstrations with varying optimality. The authors propose a framework to jointly learn confidence scores for demonstrations and a well-performing policy.

The reviewers find the research problem interesting. The strengths include: A bi-level optimization formulation of the problem, promising theoretical results, and strong empirical results. There is some concern on the applicability of the proposed method to high-dimensional complex tasks.  However, there is a clear consensus among the reviewers that the paper should be accepted.

It is recommended that the authors consider if there is a connection between this work and Zhang, et al . Causal imitation learning with unobserved confounders, NeuRIPS 2020.